# Exogenous prion-like proteins and their potential to trigger cognitive dysfunction

Jofre Seira Curto [iD][1], Adan Dominguez Martinez [iD][1,2], Genis Perez Collell[1], Estrella Barniol Simon [iD][1], Marina Romero Ruiz [iD][1], Berta Franco Bordés [iD][1], Paula Sotillo Sotillo[1], Sandra Villegas Hernandez[1], Maria Rosario Fernandez [iD][1✉] & Natalia Sanchez de Groot [iD][1✉]

## Abstract

The gut is exposed to a wide range of proteins, including ingested proteins and those produced by the resident microbiota. While ingested prion-like proteins can propagate across species, their implications for disease development remain largely unknown. Here, we apply a multidisciplinary approach to examine the relationship between the biophysical properties of exogenous prion-like proteins and the phenotypic consequences of ingesting them. Through computational analysis of gut bacterial proteins, we identified an enrichment of prion-like sequences in *Helicobacter pylori*. Based on these findings, we rationally designed a set of synthetic prion-like sequences that form amyloid fibrils, interfere with amyloid-beta-peptide aggregation, and trigger prion propagation when introduced in the yeast Sup35 model. When *C. elegans* were fed bacteria expressing these prion-like proteins, they lost associative memory and exhibited increased lipid oxidation. These data suggest a link between memory impairment, the conformational state of aggregates, and oxidative stress. Overall, this work supports gut microbiota as a reservoir of exogenous prion-like sequences, especially *H. pylori*, and the gut as an entry point for molecules capable of triggering cognitive dysfunction.

**Keywords** Microbiome; Neurodegeneration; Prion Protein; Aggregation; Amyloid

**Subject Categories** Digestive System; Microbiology, Virology & Host Pathogen Interaction; Neuroscience

## Introduction

Amyloid-forming proteins can exhibit prion-like properties, facilitating the transmission of their aggregated form between cells. A phenomenon observed in Alzheimer's disease (AD) and Parkinson's disease (PD), where amyloid-associated proteins propagate within the neuronal system of patients (amyloid-β-peptide (Aβ) and tau for AD, and alpha-synuclein for PD) (Banerjee et al, 2024; Duyckaerts et al, 2019). Moreover, this is a global event that can occur between species (Ritchie and Barria, 2021). Hence, it is of great interest to explore the structural and sequential properties that define how amyloid proteins trigger aggregation or cross-seed non-orthologous polypeptides (Ritchie and Barria, 2021; Sampson, 2025).

Prion-like proteins are polypeptides that can adopt different conformations thanks to possessing large, disordered regions. It has also been reported that these regions can contain a segment capable of driving the aggregation of the entire protein, which is the basis for the amyloid stretch hypothesis (Sabate et al, 2015; Pallarès et al, 2016; Esteras-Chopo et al, 2005). When isolated, this segment, known as the amyloid core, can not only self-assemble but also trigger the aggregation of other prion-like sequences (Krishnan and Lindquist, 2005; Osherovich et al, 2004). These core sequences are rich in amino acids that despite being associated with disordered conformations also keep amyloid propensity such as asparagine (N), glutamine (Q), and tyrosine (Y). This enhanced propensity to aggregate within a disordered segment acts as a nucleation point that favors an ordered self-assembly without the requirement of conformational unfolding (Sabate et al, 2015). Moreover, the presence and strength of these amyloid cores influence the prionogenic potential of the protein that carries them (Osherovich et al, 2004).

The origin of many neurodegenerative diseases such as AD is sporadic, and thus associated with environment and lifestyle. Recent insights have implicated both diet (Grant and Blake, 2023; Jansens et al, 2019; Lambrecht et al, 2019) and gut microbiota as potentially pivotal factors in the onset and progression of these diseases (Walker and Czyz, 2023; Walker et al, 2021; Fang et al, 2020; Hashim and Makpol, 2022). In this line, antibiotic treatments and microbiota transplantation have been raised as plausible strategies to modulate pathology (Sampson et al, 2016).

The brain-gut microbiota axis is believed to act as a bidirectional link between the gastrointestinal (GI) tract and the central nervous system. In the gut, extracellular proteins are accessible for interactions with host molecules and are also susceptible to degradation (Holmqvist et al, 2014; Kim et al, 2019b), especially in the small intestine, thus facilitating the exposure of aggregation-prone segments. Gut microorganisms and their products can be found in the brain and the vagus nerve serves to transport bacteria

[1]Unitat de Bioquímica, Departament de Bioquímica i Biologia Molecular, Universitat Autònoma de Barcelona, Barcelona, Spain. [2]Institut de Neurociències, Universitat Autònoma de Barcelona, Bellaterra, Barcelona 08193, Spain. ✉E-mail: rosario.fernandez@uab.cat; natalia.sanchez@uab.cat

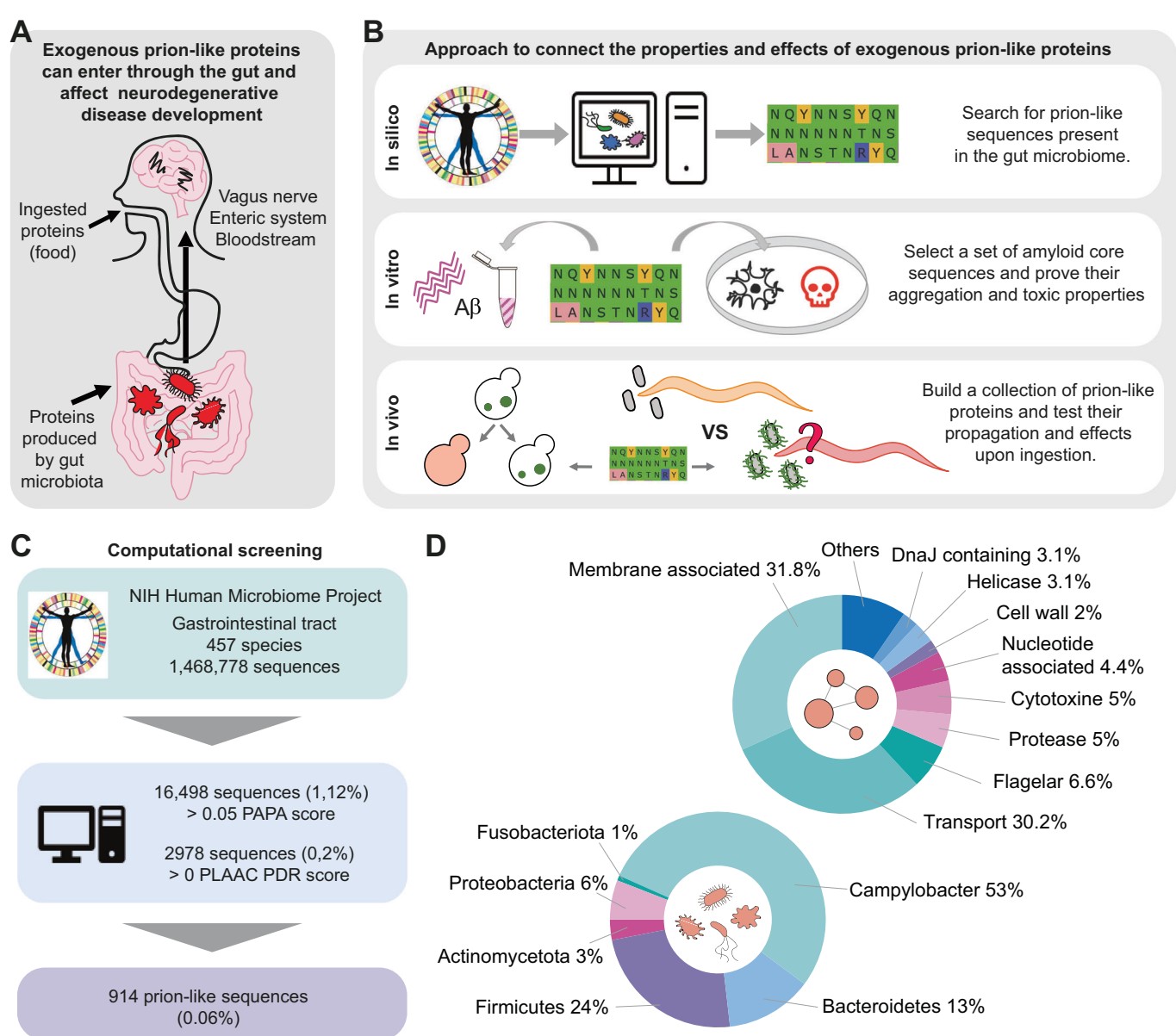

**Figure 1. Studying exogenous prion-like proteins.**

(A) The gastrointestinal tract is exposed to a diverse array of prion-like proteins, some of which are ingested while others may be produced by the resident microbiota. These proteins and their fragments can traverse the enteric system, the vagus nerve, or the bloodstream to reach the brain. (B) Diagram illustrating the approach employed in this study to investigate the properties and impacts of exogenous prion-like proteins. Top: Initial identification of prion-like sequences within the gut microbiome. Middle: Validation of computational predictions through the selection of amyloid core sequences for assessment of aggregation and toxicity. Bottom: Utilization of validated amyloid cores to construct exogenous prion-like proteins for testing their propagation and effects following ingestion in *C. elegans*. (C) Flowchart illustrating the procedure for screening of prion-like sequences in the gut microbiome and selecting amyloid-forming core candidates. (D) Top: Pie chart depicting the distribution of annotated descriptions within the 914 prion-like sequences identified in the gut microbiome. It just shows those annotations found with at least 2% of frequency. Bottom: Pie chart illustrating the distribution of phyla associated with the 914 prion-like sequences detected in the gut microbiome. It just shows those phyla found with at least 1% of frequency. Information about the genera within each phylum and its enrichment/depletion with respect to the original list is presented in Dataset EV2.

and their metabolites from the GI tract to the brain (Fig. 1A), thereby supporting their potential impact on the development and progression of neurodegenerative diseases (Bauer et al, 2016; Bonaz et al, 2018; Kim et al, 2021; Vidal-Veuthey et al, 2022; Kim et al, 2019a; Holmqvist et al, 2014; Walker and Czyz, 2023; Ritchie and Barria, 2021; Friedland and Chapman, 2017; Thapa et al, 2023).

One possible explanation of this connection is the ability of microbiota amyloid proteins to cross-seed the aggregation of host proteins. Supporting this idea, curli extracellular amyloid aggregates formed by *Escherichia coli* have been shown to accelerate alpha-synuclein aggregation in PD animal models (Walker et al, 2022; Wang et al, 2021; Sampson et al, 2020). Worryingly,

numerous homologs of the curli amyloid protein (CsgA) are found across the prokaryotic kingdom, each exhibiting different effects on alpha-synuclein aggregation (Bhoite et al, 2022; Fernández-Calvet et al, 2024). Despite this, the link between the pathologic phenotype, toxic mechanism, and molecular properties of amyloid proteins remains elusive.

To shed light on this question, our strategy integrates multi-disciplinary experiments from in silico sequence studies to phenotypical analyses in animal models. This approach facilitates the understanding of the connection between biophysical properties and the ability to trigger neurodegenerative diseases (Fig. 1B). Our computational work detected sequences with prion-like properties in 63% of the species classified as from the GI tract in an NIH Human Microbiome Project (NIH-HMP) (Peterson et al, 2009; Park et al, 2019) dataset, with a special enrichment of prion-like sequences in *Helicobacter pylori*. The amyloid cores of these sequences were rationally applied in a series of interconnected in vitro, and in vivo analyses. We designed a set of synthetic prion-like proteins to be introduced into the *C. elegans* digestive tract, which led to sensory memory impairment and lipid oxidation. The formation of immature and more reactive protein aggregates (He et al, 2012; Nimmrich et al, 2008; Lesné et al, 2013) appears to be associated with this. Overall, our findings offer insights into the relationship between the ingestion of exogenous prion-like sequences and neurodegenerative diseases, providing knowledge that may be valuable for the prevention and treatment of these diseases.

# Results

## Screening for prion-like sequences in the gut microbiome

Previous publications have reported an average of 0.3% prion-like sequences per genome in the bacterial domain (Seira Curto et al, 2022; Iglesias et al, 2015; Espinosa Angarica et al, 2013; Harrison, 2019; Lancaster et al, 2014; Toombs et al, 2012; Sabate et al, 2015; Gil-Garcia et al, 2021). To investigate their prevalence within the gut microbiome, we screened the protein sequences collected in the NIH Human Microbiome Project (NIH-HMP) (Peterson et al, 2009; Park et al, 2019) using three different algorithms: PAPA (prion aggregation prediction algorithm, https://combi.cs.colostate.edu/supplements/papa/) (Toombs et al, 2012), PLAAC (prion-like amino acid composition, http://plaac.wi.mit.edu) (Lancaster et al, 2014), and pWALTZ (Fig. 1C, https://bioinf.uab.es/pWALTZ) (Sabate et al, 2015). These approaches identify disordered sequences with a prion-like composition and provide a score that reflects the probability of a sequence behaving as a prion.

The original list derived from the NIH-HMP (http://downloads.hmpdacc.org/data/reference_genomes/body_sites/Gastrointestinal_tract.pep.fsa) (Peterson et al, 2009; Park et al, 2019) contained 457 species and 2,540,637 sequences, from which 1,468,784 are unique entries (Fig. 1C; Appendix Fig. S1). Using PAPA, we obtained 16,498 sequences (1.12%) with a positive prion aggregation propensity and when using a more stringent PLAAC score, we obtained 2978 sequences (0.20%) that had a prion-like domain. Merging these two sets of data, we identified 914 sequences (0.06%) that met both criteria, which we refer to as the positive

prion-like set (Dataset EV1). The proportion of prion-like sequences obtained with our approach (0.06%) aligns with previous reports of prion-like sequences in the bacterial domain (Seira Curto et al, 2022; Iglesias et al, 2015; Espinosa Angarica et al, 2013; Harrison, 2019; Lancaster et al, 2014; Toombs et al, 2012; Sabate et al, 2015; Gil-Garcia et al, 2021). These 914 sequences are found in 284 species, corresponding to 63% of the species collected in the GI NIH-HMP dataset. Given the ability of prion-like proteins to spread across species, this finding supports the hypothesis of gut microbiota being a reservoir of proteins with infectious potential (Seira Curto et al, 2022).

## Functional and taxonomic analysis of the prion-like sequences

One striking feature of the 914 positive prion-like sequences is the high proportion of hypothetical or uncharacterized proteins, which constitute 40% of the set (Peterson et al, 2009; Park et al, 2019) (Dataset EV1). Most of the sequences collected in the NIH-HMP have been identified via genome shotgun sequencing, and those identified as hypothetical are open reading frames without a characterized homolog in the protein databases. By examination of the disordered regions predicted by MobiDB-lite (UniProt) (Peterson et al, 2009; Park et al, 2019), it is possible to detect repeated patterns (Appendix Fig. S1 and Dataset EV1), which suggests that these proteins may have conserved yet unidentified cellular roles that require a prion-like composition (Visconti et al, 2019; Ijaq et al, 2015; Desler et al, 2012).

The remaining 60% of positive prion-like sequences, with estimated homology, can be grouped into 10 categories according to their annotated descriptions (Fig. 1D and Dataset EV1). The two main categories are membrane-associated (31.8%) and transporter (30.2%) proteins, which play direct roles in facilitating interactions between bacterial cells and the extracellular milieu.

The sequences collected in the NIH-HMP GI set are associated with 457 bacteria species. A preliminary analysis of the 914 positive prion-like sequences revealed that most belong to species whose phyla have been previously related to neurodegenerative pathologies, particularly AD (Khaled et al, 2023; Panza et al, 2019; Vogt et al, 2017). However, for a global vision of their presence, we analyzed how the positive prion-like sequences are distributed within the different phyla and compared them to the original GI NIH-HMP set (Fig. 1D and Dataset EV2).

Half of the prion-like sequences correspond to Campylobacter (53%; Fig. 1D). In our list, this phylum is predominantly represented by *Helicobacter pylori* species, and the *Helicobacter* genus presents a seven-fold enrichment compared to the sequences collected in the GI NIH-HMP set (Dataset EV2). The second most abundant phylum in terms of prion-like sequences was Firmicutes, which accounts for 24% of the identified sequences. However, it is important to note that these high quantities may be due to their abundance in the former list (48%; Dataset EV2). This phylum includes genera such as *Coprobacillus* (6%), *Eubacterium* (6.5%), *Ruminococcus* (6.5%), *Lachnospiraceae* (6.5%), *Lactobacillus* (19%), and *Clostridium* (22%), collectively representing the predominant members. 13% of the prion-like sequences come from the Bacteroidetes phylum, which is one of the most prevalent phyla in the human gut microbiome (Khiabani et al, 2023). The last phylum with more than 5% of prion-like sequences in the positive

list is Proteobacteria (Pseudomonadota), which includes several pathogenic bacteria (such as *Acinetobacter* or *Neisseria*) and the widely studied *E. coli*. However, in proportion, there are three times fewer sequences from this phylum in the prion-like positive list than in the GI NIH-HMP set.

## Computationally selected amyloid cores self-nucleate into amyloid-like fibrils

To validate the screening and to design a collection of exogenous prion-like sequences, we selected the amyloid cores of 10 prion-like positive sequences (Methods, Figs. 1B and EV1–2 and Dataset EV3). For this aim, we used pWALTZ, an algorithm that identifies the 21-amino-acid segment, within a Q/N-rich disordered region, with the highest amyloidogenic potential (Sabate et al, 2015). This length is sufficient for the formation of transmissible β-folds of the HET-s yeast prion domain (Wan and Stubbs, 2014; Wasmer et al, 2010) and is the minimal size that maximizes the discrimination between prionic and non-prionic sequences (Sabate et al, 2015).

To ensure sufficient amyloid propensity for the experimental assays, the amyloid cores were selected based on defined criteria, including pWALTZ scores above 73.55, Q/N-rich content, and host-interaction implications (see Methods). These sequences represent a diverse set of protein functions enriched in prion-like characteristics (Table EV1). The selected amyloid cores (see Methods) were named using a prefix corresponding to their species of origin, followed by a number from 1 to 10, reflecting their pWALTZ score and grouping by protein description (Table EV1, Dataset EV3). Their polypeptide sequences were chemically synthesized, and their ability to self-assemble into fibrillar aggregates was tested. The synthetic peptides were diluted in the aggregation buffer and then incubated at 37 °C without agitation. The presence of amyloid aggregates was assessed using transmission electron microscopy (TEM), Fourier-transform infrared spectroscopy (FT-IR), and binding to the amyloid dyes thioflavin-T (ThT) and Congo red (CR). All peptides exhibited the ability to form fibrillar structures (Fig. 2A) that were rich in beta-sheet conformations (Fig. 2B). Additionally, all peptides exhibited positive binding for at least one of the amyloid-specific dyes (Fig. 2C,D; Appendix Fig. S2). Together, these findings demonstrate the capacity of the predicted amyloid cores to catalyze self-nucleation into amyloid fibrils.

A set of eight control sequences with low aggregation propensity was designed to demonstrate that the observed amyloid properties are intrinsic to the selected prion-like sequences rather than artifacts (Methods). Five of these negative control sequences were derived from amyloid cores with low pWALTZ scores, while the remaining three were generated with a random amino acid composition based on the Swiss-Prot database. Using TEM, Thioflavin-T, and Congo Red we confirmed that these sequences did not self-assemble into amyloid-like aggregates (Appendix Figs. S3–5).

## Interference with host protein aggregation

We next investigated the cytotoxic properties of candidate sequences to understand how exogenous prion-like proteins might interfere with host protein aggregation. Considering their possible alteration in AD patients (Vogt et al, 2017), we examined their capacity to interfere with the aggregation of the amyloid-β peptide (Aβ).

Pre-aggregated peptides were incubated with soluble Aβ40 (Seira Curto et al, 2023) and the aggregation kinetics were monitored following the ThT fluorescence intensity (Fig. 3A; Appendix Fig. S6). The half-life (Fig. 3B) and lag times (Fig. 3C) exhibited a significant influence in 9 out of 10 peptides tested. Most peptides accelerated the aggregation kinetics, except for HA10, which slowed it down (Fig. 3A–C; Appendix Fig. S6).

In the analysis of the relationship between the peptide's composition and its effect on Aβ40 aggregation, there was a robust correlation between the peptide net charge and lag-time kinetics, except for RI6 (Fig. 3D). A more positive net charge results in faster Aβ40 aggregation, whereas a more negative charge results in slower aggregation. This effect may be attributed to electrostatic repulsion forces. Under the experimental conditions used (pH 7.4 and 100 mM NaCl), Aβ40 carries a negative charge (−2.9) (Seira Curto et al, 2023; Wood et al, 1996). As a result, positively charged peptides may neutralize the net charge favoring Aβ40 assembly, while negatively charged peptides may enhance the repulsive forces present in the mixture. This has also been observed in screening for Aβ aggregation inhibitors (Liao et al, 2012; Chan et al, 2012; Liu et al, 2016). Moreover, these works also report that these electrostatic forces work similarly for Aβ42 (the main isoform located in the amyloid plaques), which despite having two extra amino acids have the same net charge at physiological conditions.

To test whether the full protein, and not just the amyloid-forming core, can form fibrillar aggregates and influence Aβ40 aggregation kinetics, the parental protein that contains the BH4 amyloid core (C9L6N5, a protein with a DNAJ domain) was also tested. This protein, composed of 212 amino acids, was expressed in *E. coli* and purified (Appendix Fig. S7). The purified protein was able to aggregate into fibrillar structures (Appendix Fig. S7). Subsequent incubation of Aβ40 with pre-aggregated C9L6N5 accelerated the aggregation kinetics similarly to the amyloid-forming core (Appendix Fig. S7). This result supports that not only the amyloid-forming cores but also the proteins from which these sequences originate can form fibrillar aggregates (Batlle et al, 2017; Osherovich et al, 2004; Pallarès et al, 2018) and seed host molecules.

## Toxicity and oxidative stress in neuron-differentiated cells

We next assessed whether the amyloid core peptides derived from bacteria residing in the gut could induce phenotypic hallmarks associated with neurodegenerative diseases in neuronal differentiated (SH-SY5Y) cells. We first analyzed whether the aggregates maintain their β-sheet conformation under cell culture conditions using an FTIR microscope (µFTIR), a technique employed to measure the presence of amyloid aggregates in cells (Benseny-Cases et al, 2018) (Methods, Appendix Figs. S8–9). Then, we evaluated their potential to induce cell death using the MTT assay and oxidative stress using the DCFDA/H2DCFDA assay (Royall and Ischiropoulos, 1993) (Fig. 3E,F; Appendix Fig. S9).

The MTT assays showed that peptides with a net charge close to 0 exhibited greater toxicity compared to those with a higher charge (Fig. 3G). This suggests that electrostatic interactions might play a protective role against cytotoxicity. While differing from the Aβ results, where positively charged peptides accelerated the aggregation kinetics (Fig. 3D), this also indicates the existence of additional mechanisms contributing to toxicity.

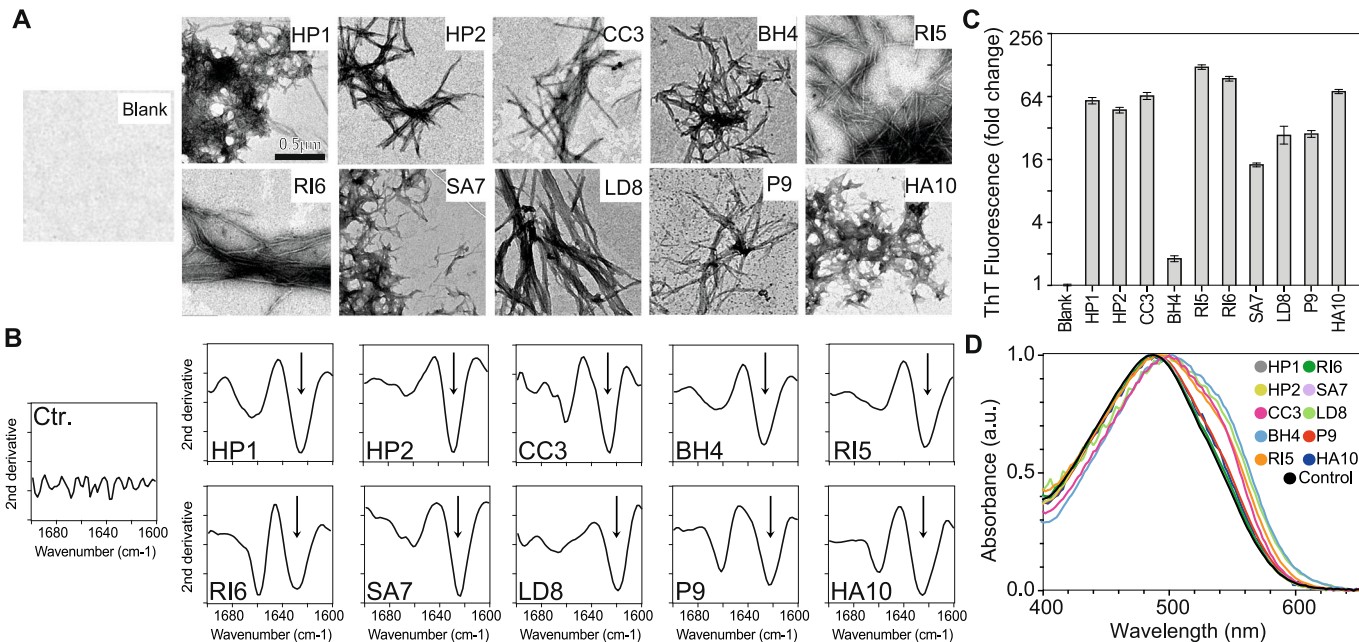

**Figure 2. Amyloid aggregation of sequences from bacteria residing in the gut.**

(A) TEM images showing fibrillar aggregates formed by the selected amyloid cores. (B) FTIR spectrum of the aggregated peptides. The arrows indicate the b-sheet intermolecular signal. (C) Thioflavin-T fluorescence increase (fold change) upon binding the aggregated peptides. All samples with aggregated peptides are significantly more fluorescent than Th-T alone. Statistical analysis was performed using multiple unpaired two-tailed t-tests, corrected for multiple comparisons using the Benjamini–Hochberg method (FDR set at 5%). The exact statistical values are provided in Dataset EV4. (D) Congo Red absorbance in the presence of the aggregate's peptides (Appendix Fig. S2 for individual scans). Control is buffer with CR, without aggregates. All experiments (from (A) to (D)) were performed with three biological replicates ($N = 3$), each consisting of three technical replicates. Source data are available online for this figure.

Polar, uncharged polypeptides tend to exhibit amphipathic characteristics that facilitate their interaction with membranes. Similar to antimicrobial peptides, these interactions can lead to membrane disruption and subsequent cell toxicity (Elliott et al, 2020). The observed general drop in cell viability to below 70% (Fig. 3E) suggests that the selected amyloid cores from the gut microbiome can induce cellular damage upon aggregation.

The production of cellular reactive oxygen species (ROS), significantly increased in six out of ten peptides compared to the control without aggregates (Fig. 3F). These data suggest that the reduced cell viability (Fig. 3E) can, in part, be attributed to oxidative stress. Overall, our results suggest that prion-like aggregates can cause cellular damage through membrane interactions and ROS production (De Groot and Burgas, 2015; Auten and Davis, 2009), supporting their potential contribution to disease (Fig. 3H).

Among all the peptides tested, the aggregates of HP1 and HP2 exhibited especially high toxicity (Fig. 3E; Appendix Fig. S10) and reduced cell viability to nearly 30% at 10 μM. These two amyloid cores may be important sequences for the oligomerization and toxic potential of their parental proteins, two putative vacuolating cytotoxins (Dataset EV3).

## Bacterial amyloid cores from the gut are functional in yeast

Although all the core sequences formed amyloid fibrils and exhibited cytotoxicity, each sequence has distinct properties, which

were reflected in their different ways of displaying these traits (Figs. 2 and 3). This combined with the amyloid stretch hypothesis, which states that the amyloidogenicity of a protein is comprised in short protein stretches (Esteras-Chopo et al, 2005), leads us to hypothesize that incorporating these amyloid cores into the same prion-like framework would result in a series of chimeras with different prion properties (Zambrano et al, 2015; Sabate et al, 2015; Osherovich et al, 2004; Von Der Haar et al, 2007).

The exchange of prion-like domains is commonly studied in the yeast prion Sup35 since it allows monitoring protein aggregation and propagation easily through an in vivo nonsense suppression assay. Additionally, Sup35 has been shown to accelerate the aggregation of tau and alpha-synuclein proteins in AD and PD models, respectively (Meng et al, 2023; Flach et al, 2022). These effects were demonstrated through intrahippocampal inoculation of Sup35NM fibrils in P301S tau transgenic mice, and through nasal infection with *Saccharomyces cerevisiae* in α-syn A53T transgenic mice.

The residues 7–13 recapitulate the amyloid properties of the full-length Sup35 (Burra et al, 2021), and the deletion of the first 40 residues, known as the nucleation domain, destroys the prion phenotype [PSI+] (Balbirnie et al, 2001; Krishnan and Lindquist, 2005). Based on this, we removed the first 40 amino acids of the yeast prion Sup35 and replaced them with our collection of amyloid cores (Fig. 4A; Appendix Supplementary Data) (Parham et al, 2001; Wickner et al, 2015). The sequential analysis of the resultant chimeras shows similar aggregation and prion-like propensities as the original Sup35 (Appendix Supplementary Data). This indicates

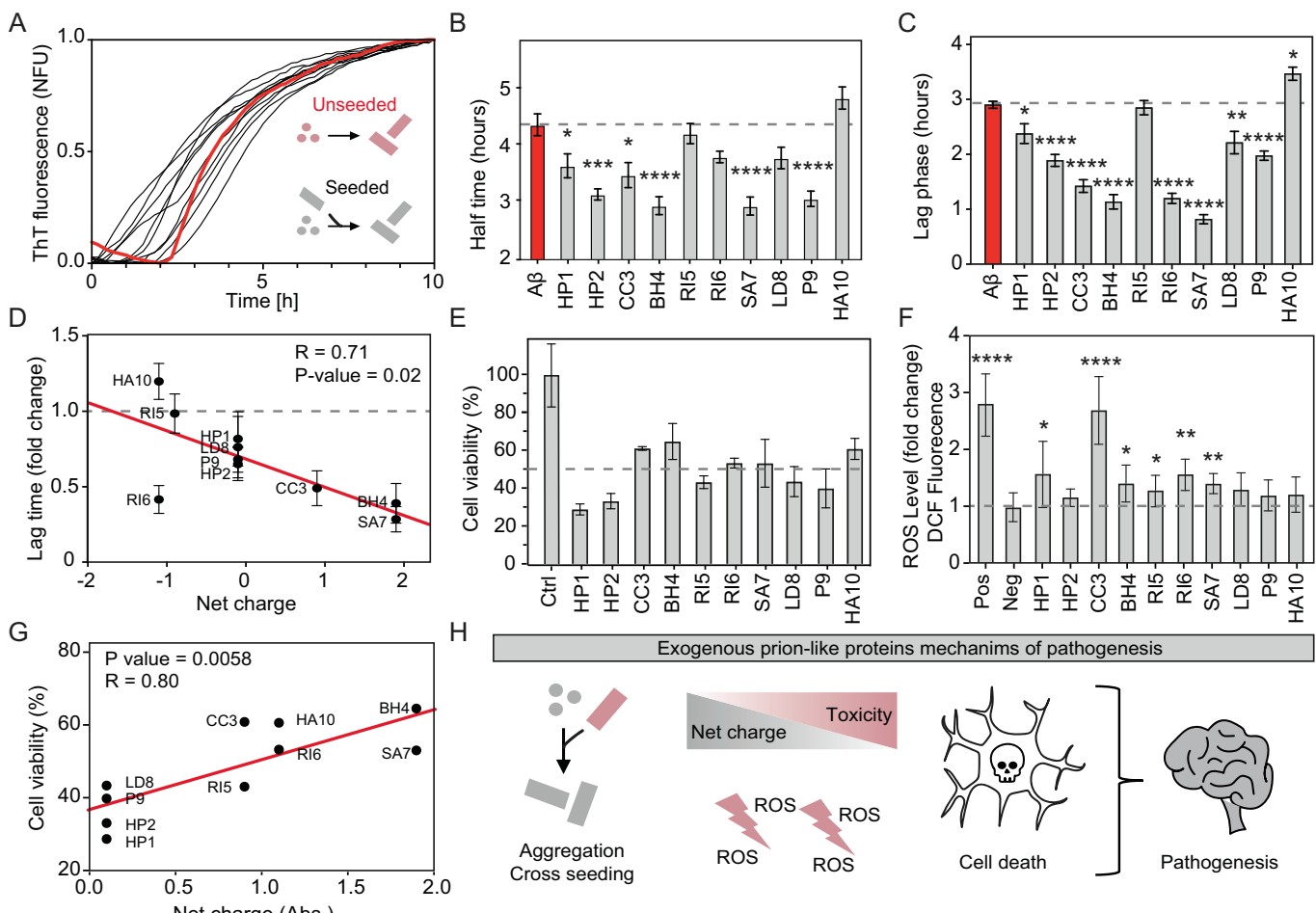

**Figure 3. Seeding and toxicity potential of sequences from the gut microbiome.**

(A) Aggregation kinetics of Aβ40 peptide seeded by amyloid-forming cores from the gut microbiome (for individual kinetics, see Appendix Fig. S6). The kinetic assays were conducted with four biological replicates, each with three technical replicates. (B) Half-times of seeded aggregation kinetics (four biological replicates, each with three technical replicates). Error bars indicate SEM. In all panels (B, C, E, F), significance relative to the appropriate control (Aβ, Ctrl, or Neg) was assessed using multiple unpaired two-tailed t-tests, corrected for multiple comparisons using the Benjamini–Hochberg method (FDR set at 5%; *$q < 0.05$, **$q < 0.01$, ***$q < 0.001$, ****$q < 0.0001$). The exact statistical values are provided in Dataset EV4. (C) Lag phases of seeded aggregation kinetics (four biological replicates, each with three technical replicates). Error bars indicate SEM. (D) Linear correlation between lag time and the net charge of the amyloid-forming cores used for seeding. The plot also shows the Pearson correlation coefficient (R) and the p-value calculated from an F-test. Without RI6, R increases to 0.95. (E) Viability of neuron-differentiated SH-SY5Y cells assessed by MTT assay after incubation with 10 μM peptide aggregates. All cells incubated with aggregated peptides showed significantly lower viability (FDR 5%) than control cells (Ctrl) without peptides. Error bars indicate SD, centered on the mean. Three biological replicates ($N = 3$) were conducted, each consisting of three technical replicates. For results at lower concentrations, see Appendix Fig. S10. (F) DCFDA/H2DCFDA cellular ROS assay. The first column (Pos) shows the signal from the positive control (TBHP). Fluorescence was measured after 4 h of exposure to the peptides. Bars show the fluorescence fold change relative to the negative control (Neg). Error bars indicate SD, centered on the mean. Three biological replicates ($N = 3$) were conducted, each consisting of three technical replicates. (G) Linear correlation between peptide net charge and cell viability after treatment with 10 μM peptide aggregates. The plot also shows the Pearson correlation coefficient (R) and the p-value calculated from an F-test. (H) Schematic representation of the proposed role of aggregation in pathogenesis. Source data are available online for this figure.

that the bacterial-derived amyloid cores can functionally substitute the Sup35 nucleation domain, as the new proteins retain the ability to aggregate and propagate, supporting their role as exogenous prion-like proteins.

We used the nonsense suppression assay to evaluate the formation and propagation of the Sup35 aggregates ([*PSI*⁺]) (Parham et al, 2001), including those variants incorporating bacterial-derived amyloid cores (see Methods). As positive and negative controls, we employed cells expressing unmodified Sup35 prion sequence (Sup35NM) and Sup35 lacking the first forty amino acids (ΔSup35), respectively. Microscopy analysis revealed that

Sup35NM concentrates the fluorescence in bright foci, while ΔSup35 exhibits a homogeneous distribution throughout the cytosol, indicating aggregation and solubility, respectively (Fig. 4B and Dataset EV5). In agreement with the computational prediction (Appendix Supplementary Data), all the cells expressing a Sup35 variant carrying an amyloid core (from HP1 to HA10) exhibited fluorescent foci (Fig. 4B).

Consistently, the [*PSI*⁺] prion propagation assay displayed 100% white colonies in Sup35NM, dropping to 30% in ΔSup35 (Fig. 4C,D). In the case of cells expressing a Sup35 variant, despite presenting a high [*PSI*⁺] diversity, all showed a significant increase

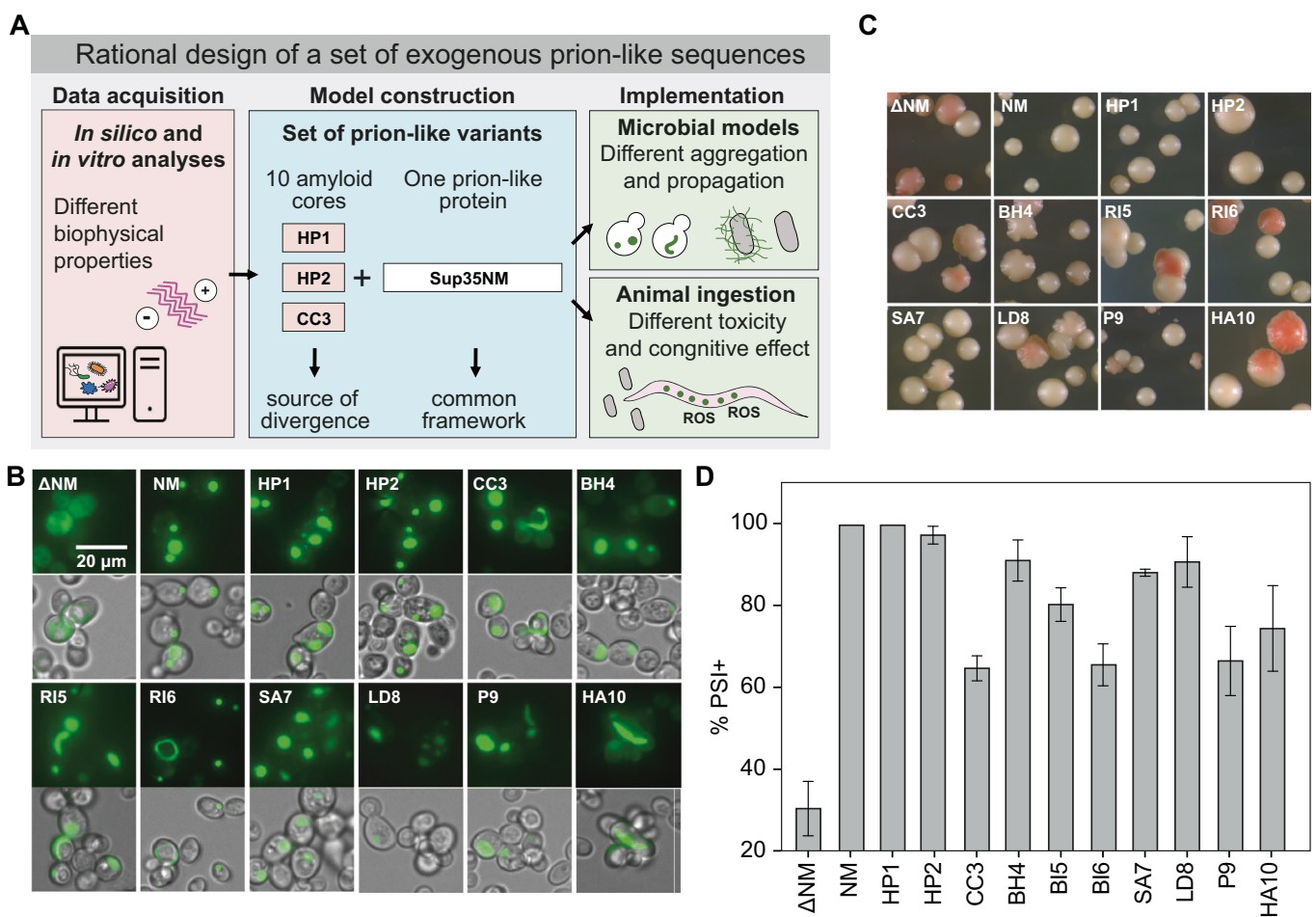

**Figure 4. Aggregation and propagation analysis in yeast.**

(A) Diagram illustrating how we designed the set of exogenous prion-like sequences studied in this work. Left panel: We first looked for 10 different amyloid-cores within the gut microbiome and analyzed their potential to trigger amyloid aggregation. Medium panel: The amyloid cores were introduced into the same protein (Sup35) to create 10 chimeras with different prionic properties. Right panel: Their aggregation and propagation were tested in vivo in yeast and bacteria models. Finally, the chimeras, expressed by bacteria, were studied as exogenous prion-like proteins that are ingested by *C. elegans*. (B) Fluorescent images showing intracellular aggregates of the Sup35-GFP variants. (C) Representative images of white/red colonies from the non-sense suppression assay. (D) Percentage of [PSI+] (prionic behavior) for each yeast strain analyzed. All assays were performed in three biological triplicates ($N = 3$). Error bars indicate SD. Yeast strains expressing Sup35 chimeras with the predicted amyloid cores exhibited significantly higher [PSI+] levels (indicating better prion propagation) compared to the strain without a prion core (ΔNM). Statistical significance was assessed using multiple unpaired two-tailed t-tests, corrected for multiple comparisons using the Benjamini–Hochberg method (FDR set at 5%; all samples resulted in $q < 0.05$). The exact statistical values are provided in Dataset EV4. Source data are available online for this figure.

in the number of white colonies ([*PSI*⁺]) compared to ΔSup35 (Fig. 4D).

Overall, both assays underscore the significance of the nucleation domain in driving Sup35 aggregation and propagation. The data also show that all the bacterial protein fragments selected can trigger Sup35 nucleation and can replicate its prion transmission in vivo in yeast. It is worth noting that this result is not obvious, since only prion-like and amyloidogenic sequences can restore the Sup35 aggregation and propagation abilities (Alberti et al, 2009; Zambrano et al, 2015; Sabate et al, 2015; Osherovich et al, 2004; Cascarina et al, 2017).

A deeper analysis of the microscopy images shows differences in the colonies and the aggregated forms between Sup35 variants. Importantly, the color stability of the [*PSI*⁺] colonies is indicative of

the prion-like protein capacity to keep the aggregated conformation through different generations. We observed that some Sup35 chimeras form long ring intracellular structures (Fig. 4B, Dataset EV5), and that the presence of more than 10% of cells with ring forms is associated with the formation of color-revertant colonies (indicative of a low-stability aggregated conformation). These ring forms are large, early aggregated stages that can progress to a mature single punctate conformation through the fragmentation activity of the HSP104 chaperone (Tyedmers et al, 2010; Sharma et al, 2017). In line with our results, the presence of long intracellular fibers has also been associated with less stable [*PSI*⁺], possibly due to lower amounts of prion seeds (propagons) small enough to propagate between cells; meanwhile, the punctate focus is associated with the formation of efficiently transmissible [*PSI*⁺]

(Chernova et al, 2017; Liebman and Chernoff, 2012; Derkatch and Liebman, 2013). Among all the prion-like variants, BH4 was the only one not presenting rings or multiple foci, indicating the formation of especially mature and stable aggregates (De Groot et al, 2015; Carija et al, 2017).

The Sup35 variants HP1, HP2, BH4, SA7, and LD8 demonstrated the highest percentage of [$PSI^+$] conversion and, like Sup35NM, are robustly stable, as is evident by the absence of revertant colonies. Of these, HP1, HP2, and LD8, also displayed multiple fluorescent punctate foci, mirroring the pattern observed in Sup35NM (Fig. 4B, Dataset EV5). Interestingly, in vitro, upon incubation with SH-SY5Y, these three sequences also led to some of the lowest cell viabilities. Similarly, recent studies on variants of TDP-43 in yeast show that the presence of multiple punctate foci corresponds to more dynamic, liquid-like deposits with enhanced toxicity (Bolognesi et al, 2019). As observed in the in vitro assay, this toxicity may be associated with the original roles that these sequences may have in the parental protein (Dataset EV2–3).

The low aggregation-prone control sequences (Appendix Fig. S11) were introduced at the N-terminal of Sup35-GFP to demonstrate that not all sequences can trigger foci assembly, as this requires a prion- or amyloid-like composition. As expected, the resultant chimeras failed to form intracellular aggregates in yeast, except for L2, where 13% of the cells exhibited foci (substantially lower than the 61% to 100% observed in strains expressing aggregation-prone bacterial sequences). Overall, the nonsense suppression assay was consistent with the sequence analyses (Appendix Supplementary Data) demonstrating that the Sup35 variants carrying the bacteria amyloid cores can aggregate and propagate in a prion-like manner. Hence, this assay went further by showing different aggregation and propagation patterns.

## Aggregation in a bacteria-based C-DAG system

With the aim of (i) simulating protein aggregation within a bacterial context and (ii) building a comestible gut-colonizing vehicle for our collection of exogenous prion-like proteins, we introduced the same set of Sup35NM chimeras expressed in yeast into the bacteria-based C-DAG system (Fig. 5A). In this system, the proteins are fused to the CsgAss exportation signal of the curli fibers (Appendix Supplementary Data), and the protein aggregation propensity can be monitored by measuring the formation of extracellular aggregates (Sivanathan and Hochschild, 2013).

To detect the formation of extracellular aggregates, we grew E. coli C-DAG cells on CR plates. The presence of extracellular aggregates is revealed by the development of reddish colonies. All strains, except for BH4 and RI5, displayed a robust signal, notably surpassing that exhibited by the soluble M domain of Sup35 (Sup35M) (Fig. 5B; Appendix Fig. S12). We also observed by TEM that all strains except Sup35M and ΔSup35 displayed abundant large and fibrillar aggregated structures surrounding the cells (Fig. 5C).

When the Sup35 variants containing the rationally designed low aggregation-prone sequences were expressed by the E. coli C-DAG cells, no extracellular aggregates were observed (Appendix Fig. S13). Similar to the results in yeast, these findings emphasize the critical role of the nucleation core in driving Sup35 aggregation and demonstrate that sequence variations lead to distinct aggregation behaviors.

## Bacterial extracellular fibrils induce loss of associative memory

Once analyzed their aggregation and propagation capabilities, we used the C-DAG system as a vehicle to introduce the exogenous prion-like proteins into the C. elegans gut to analyze their effects on the host, in particular, a neurodegenerative disease phenotype (Figs. 5A and 6A). By studying the same protein sequences across different assays, from biophysics to phenotypic analyses, we aim to combine these results to achieve a mechanistic understanding (Fig. 1B).

We start by monitoring alterations in the sensory memory of two different strains of C. elegans: the laboratory wild-type N2 (Fig. 6) and the AD model CL2355, which has a pan-neuronal expression of Aβ42 (Appendix Fig. S14). The animals were fed the E. coli C-DAG collection expressing the Sup35NM chimeras (Fig. 5A). Subsequently, we performed a short-term associative memory assay (STAM) (see Methods) that links the presence of food with the chemoattractant butanone, a volatile odorant (Fig. 6A), to assess the cognitive status of the worms (Stein and Murphy, 2014). Both C. elegans models were fed E. coli expressing either the ΔNM or NM variants of Sup35. Compared to N2, CL2355 presented lower values in both indices, learning and memory. Interestingly, N2 worms that were fed bacteria that form aggregates (Sup35NM), exhibited a memory index similar to that of CL2355 worms, which were fed bacteria unable to form extracellular aggregates (ΔSup35). Since both bacteria strains share the same genotype, these results suggest that the ingestion of the extracellular aggregates produced by Sup35NM causes cognitive impairment in the wild-type animal (N2) similar to those associated with the model of AD (CL2355). Furthermore, this assay also showed that the disease phenotype in CL2355 could be exacerbated when Sup35NM bacteria were included in its diet.

After validating the STAM assay to measure the neurodegenerative phenotype in N2, we fed the nematodes with E. coli expressing the Sup35NM variants containing the bacteria-derived amyloid cores. As a control, we first compared the non-conditioned worms (naive) and, as expected, no significant differences were observed between the animals fed with ΔSup35 and those fed with the strains capable of forming extracellular aggregates (Fig. 6B, N data points). This indicates that without prior conditioning, all the animals respond similarly to the chemoattractant. However, during the learning assay (Fig. 6B, L data points), N2 worms fed with bacteria expressing HP1, HP2, RI5, RI6, or HA10 exhibited an index value less than half of that measured for those worms fed with bacteria expressing ΔSup35 or Sup35NM. This suggests that the consumption of these bacterial strains and their extracellular aggregates has a stronger physiological impact on learning ability. Notably, the only variable among these conditions is the E. coli strain consumed and, consequently, the specific chimera it expresses.

When added to the nematodes' diet, 8 out of 10 bacteria strains triggered a significant memory decline (Fig. 6B, M data points), measured as a reduced chemotaxis index compared to ΔSup35. This indicates that most of the prion-like chimeras can form aggregates that, upon ingestion, lead to a decline in the cognitive abilities of the worm. However, some proteins had a more pronounced effect on learning (HP1, HP2, RI5, RI6, and HA10), while others predominantly influenced memory (SA7, LD8, and P9) (Stein and

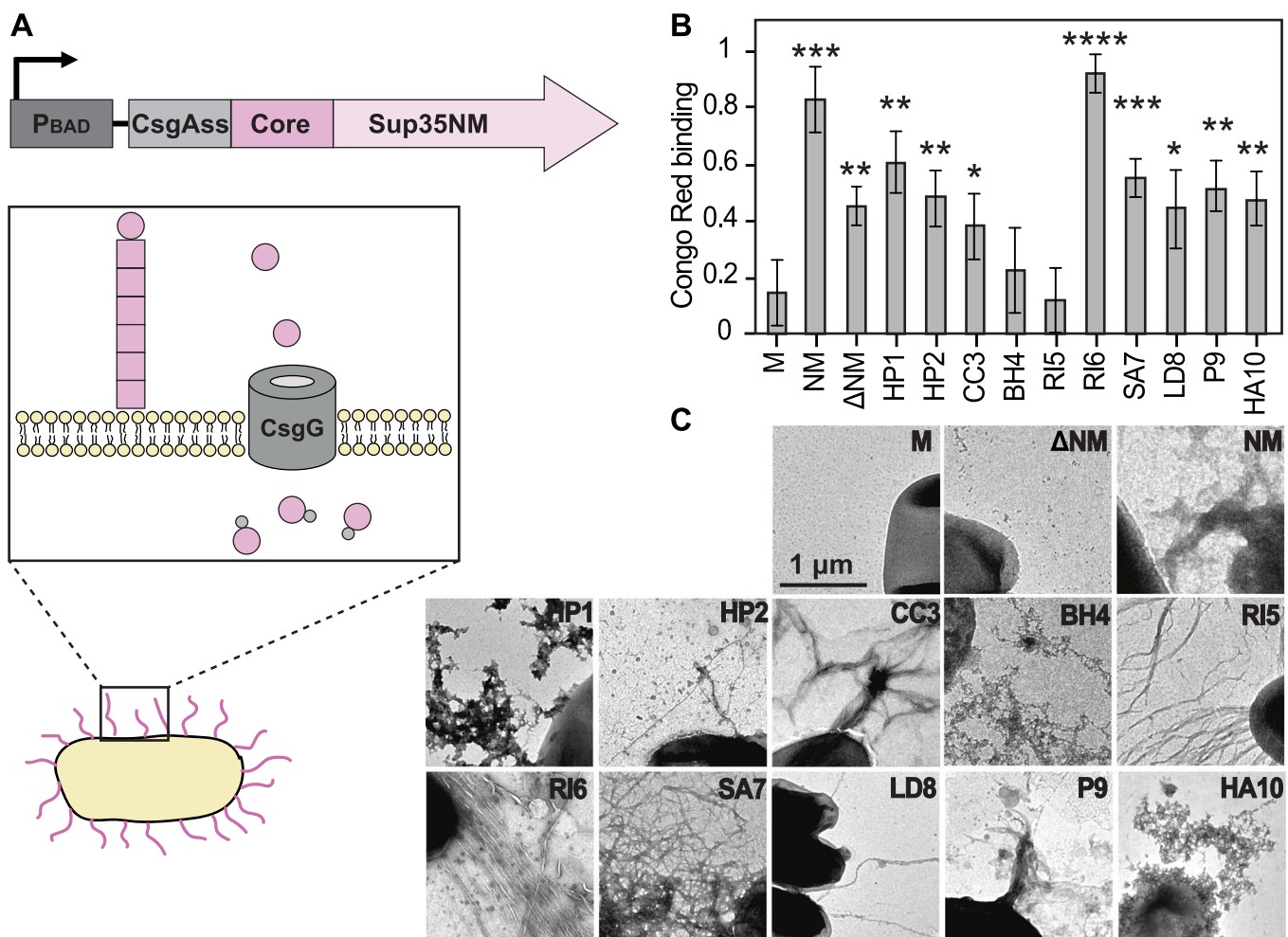

**Figure 5. Aggregation in bacteria C-DAG system.**

(A) Diagram showing the C-DAG construct integrated into *E. coli* cells and how the expression of Sup35p chimeras can form extracellular protein aggregates.
(B) Measurement of the colony red intensity using ImageJ, normalized between maximum and minimum values. Asterisks indicate the significance in red color increase compared to Sup35M, the variant encoding the soluble Sup35 medium region. Error bars indicate SD, centered on the mean ($N = 4$ technical replicates). Statistical analysis was performed using multiple unpaired two-tailed t-tests, corrected for multiple comparisons using the Benjamini–Hochberg method (FDR set at 5%; *$q < 0.05$, **$q < 0.01$, ***$q < 0.001$, ****$q < 0.0001$). The exact statistical values are provided in Dataset EV4. (C) TEM images showing fibrillar aggregated structures around *E. coli* cells. Source data are available online for this figure.

Murphy, 2014). In contrast, the ingestion of bacteria expressing three of the low aggregation-prone Sup35 controls (L1, L2, and R1) did not result in significant learning or memory deficits (Appendix Fig. S15). These analyses of cognitive abilities highlight the roles of aggregation and prion-like behavior as key factors in toxicity, underscoring the far-reaching influence of the microbiota's amyloidogenic sequences on cognitive function (Fig. 1A). Moreover, the results also point out that different prion-like aggregates can impact cognitive abilities in distinct ways (Fig. 6B).

## Ingestion of exogenous prion-like proteins increases lipid oxidation

To explore the factors underlying the worms' cognitive decline, we analyzed them using synchrotron radiation μFTIR (SR-μFTIR). This technique measures the absorption spectra of the molecular bonds without the need for labeling. It is a non-invasive method

that captures the distinct chemical signatures of cells (Baker et al, 2014) and also *C. elegans* specimens (Gonzalez-Moragas et al, 2017; Muñoz-Juan et al, 2024). Here, we utilized SR-μFTIR to analyze the lipid oxidation and the protein/lipid proportion (Fig. 6C,D; Appendix Figs. S16–18). These analyses show that, in general, the worms with better memory indices tended to present lower lipid oxidation levels together with lower protein/lipid signals (Fig. 6E). Hence, these results suggest a possible protein accumulation and an increase in ROS in the nematodes showing a cognitive decline. This agrees with the cellular analysis of the amyloid cores (Fig. 3F), which also suggested that oxidative stress could have a significant contribution to their toxicity.

To further investigate how the entry of exogenous amyloid proteins results in the accumulation and subsequent generation of ROS, we examined a nematode intestinal lysosome-related organelle known as the gut granule (Hermann et al, 2005; Dell'Angelica et al, 2000) (Fig. 7; Appendix Fig. S19). These

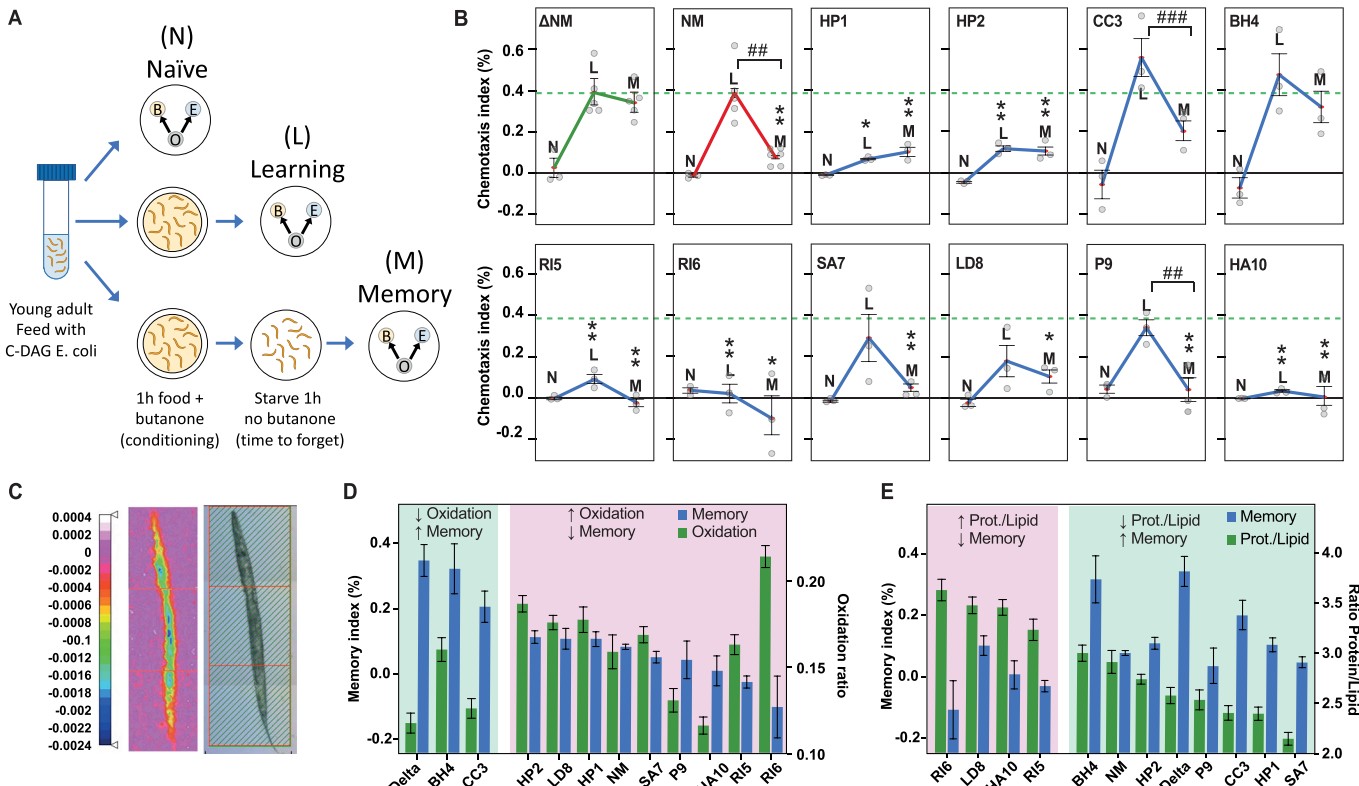

**Figure 6. Cognitive effects on *C. elegans*.**

(A) Diagram showing the three different conditions of the STAM assay. (B) Plots showing the three chemotaxis indices measured: naïve (N), learning (L), and memory (M). Each data point represents 300–500 worms. The green dashed line indicates the learning index measured with ingestion of ΔSup35 (ΔNM). Error bars represent the SEM. Asterisks (*) indicate significance relative to the corresponding ΔNM condition (Naive, Learning, or Memory), based on multiple unpaired two-tailed t-tests corrected using the FDR with the Benjamini–Hochberg method (FDR set at 5%; $N = 3$ biological replicates of 300–500 worms each; *$q < 0.05$, **$q < 0.01$, ***$q < 0.001$, ****$q < 0.0001$). The exact statistical values are provided in Dataset EV4. (#) Hash symbols indicate significance between indices of worms fed the same bacterial strain but exposed to different conditions (Learning vs. Memory). Significance was assessed using an ordinary two-way ANOVA followed by Bonferroni's multiple comparisons test ($N = 3$ biological replicates of 300–500 worms each; #$p < 0.05$, ##$p < 0.01$, ###$p < 0.001$). The exact statistical values are provided in Dataset EV4. (C) mFTIR of a N2 nematode feed with CC3 strain. Shows the distribution along the nematode body of the amyloid signal (1629/1654). For this parameter, we observed no significant increase between the worms ingesting ΔSup35 (ΔNM) and those fed with the other Sup35 variants. (D) Comparison between memory indices and whole-body oxidation ratio measured with mFTIR (error bars are SEM; for oxidation ratio $N \geq 44$, the measures were taken uniformly along the body of 10 different worms). A linear regression analysis is presented in Appendix Fig. S16. The samples are arranged from highest to lowest memory index values. The green zone indicates worms with higher memory index values, while the red zone represents those with lower memory index values. (E) Comparison between memory indices and whole-body protein/lipid ratio measured with mFTIR (error bars are SEM; for protein/lipid ratio $N \geq 42$, the measures were taken uniformly along the body of 10 different worms). A linear regression analysis is presented in Fig. S16. The samples are arranged from highest to lowest Protein/Lipid IR signal ratio. The red zone indicates worms with a higher Protein/Lipid IR signal ratio, while the green zone represents those with a lower one. Source data are available online for this figure.

organelles are one of the nematode's first defenses against the entrance of exogenous menaces through the intestine (Hajdú et al, 2023). They are involved in digestion and nutrient storage and are able to sequester and neutralize ingested toxins and pathogens to prevent damage (Roh et al, 2012; Chun et al, 2017; Ardelli and Prichard, 2013). Recently, they have also been associated with the stress response and with a reduction in protein aggregation in a Huntington's disease model (Brunquell et al, 2018).

The gut granules contain several autofluorescent compounds (Fig. 7A) whose intensity increases with oxidative stress and cellular damage (Navarro-Hortal et al, 2024; Ayuda-Durán et al, 2020). The ingestion of the *E. coli* C-DAG collection increased the number of gut granules (Appendix Fig. S19) and their fluorescence intensity (Appendix Fig. S20) compared to the consumption of the *E. coli* OP50 strain, the standard nematode food and a biofilm-defective

mutant (Arata et al, 2020). This observation suggests an increase in gut granule activity and that the products expressed by the *E. coli* C-DAG bacteria may interfere with these organelles more than those produced by the OP50 strain. In agreement with previous studies on oxidative stress (Navarro-Hortal et al, 2024; Ayuda-Durán et al, 2020), our analyses with SR-μFTIR indicate that the lipid oxidation signal in the nematode intestine is linked with the number of fluorescent foci (Fig. 7B). Overall, this result connects the aggregates produced by the ingested bacteria with the oxidative levels of the nematode and the number of gut granule organelles.

## The aggregates' properties define the cognitive deficit

We next compared the number of gut granules with the memory index, revealing that the worms with lower indices tend to have

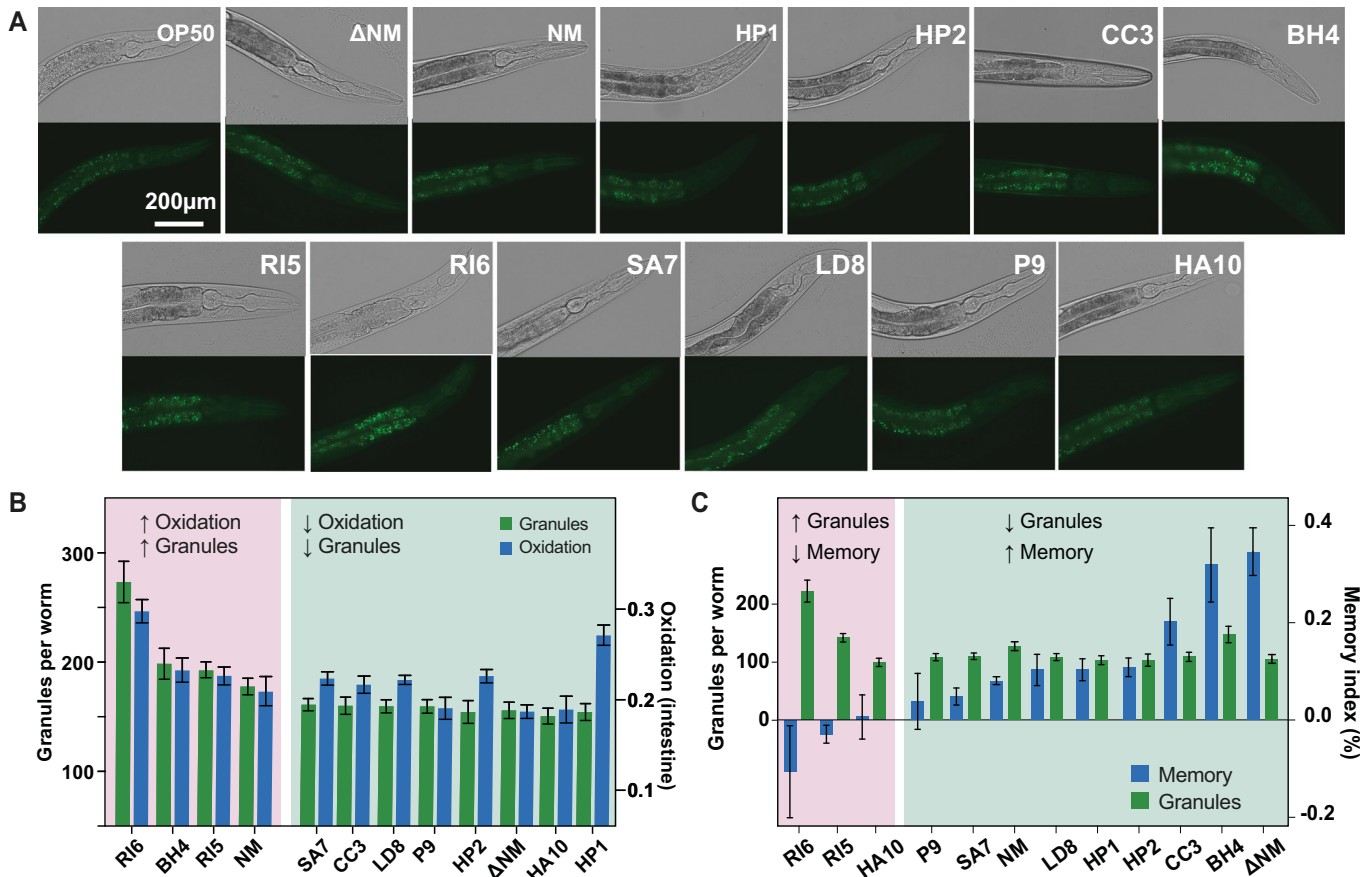

**Figure 7. Oxidative stress and gut granules fluorescence.**

(A) Fluorescent images showing the presence of gut granules in the analyzed worms. At the top transmitted light, and at the bottom fluorescence. (B) Comparison of lipid oxidation within the intestinal section and the number of gut granules ($N = 10$ different worms). Lipid oxidation and granule number were significantly different between the red group (RI6, BH4, RI5, NM) and the green group (SA7, CC3, LD8, P9, HP2, ΔNM, HA10, HP1), as determined by unpaired two-tailed t-tests ($p = 8.72 \times 10^{-7}$ for lipid oxidation and $p = 1.76 \times 10^{-6}$ for granule number). Each group's values were pooled and analyzed independently. Error bars represent the standard error of the mean (SEM). (C) Comparison of memory index ($N = 3$ biological replicates of 300–500 worms each) and the number of gut granules ($N = 10$ different worms). Memory index and granule number were significantly different between the red group (RI6, RI5, HA10) and the green group (P9, SA7, NM, LD8, HP1, HP2, CC3, BH4, ΔNM), as determined by unpaired two-tailed t-tests ($p = 0.0106$ for memory and $p = 2.45 \times 10^{-7}$ for number of granules). Each group's values were pooled and analyzed independently. Error bars represent the standard error of the mean (SEM). Source data are available online for this figure.

more gut granules (Fig. 7C; Appendix Fig. S16). This result suggests that the accumulation of components in the gut granules and the oxidation associated with the intake of *E. coli* C-DAG strains may be related to the severity of memory loss.

Similarly, the worms ingesting *E. coli* strains that formed intensely red colonies in the presence of CR tend to have a lower memory index (Appendix Fig. S21), except for RI5, which produced an especially faint red color. In addition, the proteins that result in the formation of abundant ring structures in yeast are associated with worms with lower memory indexes (Appendix Fig. S22).

Taking all the assays together, there appears to be a parallel progression in the number of gut granules, the intensity of red colonies, and the formation of ring aggregates in yeast, which seems to be linked with memory loss (Fig. 7C; Appendix Fig. S22). The intensity of red in the *E. coli* colonies is associated with the ability of the extracellular aggregates to bind CR. In yeast, the ring structures are precursors of the foci, with poorer propagation and enhanced toxicity (Chernova et al, 2017; Derkatch and Liebman,

2013; Barbitoff et al, 2022). On the contrary, the formation of large stable foci has been associated with a protective strategy against oxidative damage (Carija et al, 2017). In this line, an increase in gut granule fluorescence has been associated with an increase in lipid oxidation (Navarro-Hortal et al, 2024; Ayuda-Durán et al, 2020). Considering all factors together, we hypothesize that the inability to progress to mature and inert aggregates(He et al, 2012; Nimmrich et al, 2008; Lesné et al, 2013) may lead to the formation of promiscuous conformations (Nimmrich et al, 2008; Lesné et al, 2013) that generate more ROS and ultimately promote stronger cognitive decline.

## Discussion

The gut serves as a potential entry point for various exogenous molecules, including prions. There, the microbiota can act as a source of amyloid-promoting proteins with proven effects on host

health(Walker et al, 2022; Wang et al, 2021; Sampson et al, 2020; Bhoite et al, 2022; Fernández-Calvet et al, 2024; Seira Curto et al, 2022). To explore the impact of exogenous prion-like proteins on the host, we assembled a collection of potential amyloidogenic sequences. These sequences were computationally designed based on gut microbiome analysis, providing a synthetic and simplified approach that does not intend to represent any specific natural protein but instead aims to offer a broad overview of prion-like sequences. The resulting peptides were capable of self-assembling into amyloid-like fibrillar structures, with the ability to interfere with Aβ aggregation and induce ROS production in neuron-differentiated cells.

The identified prion-like sequences are associated with diverse functional categories, including membrane receptors, transporters, and adhesins, indicating their involvement in critical interactions with host cells and the environment (Visconti et al, 2019; Ijaq et al, 2015; Desler et al, 2012). Notably, certain bacterial proteins, like curli fibers found in *E. coli*, may enhance bacterial colonization and pathogenesis, potentially influencing neurodegenerative processes(Chen et al, 2016; Friedland et al, 2020; Van Gerven et al, 2018; Kuwajima et al, 2022; Kosolapova et al, 2020; Wang et al, 2021).

Among the bacteria species detected, *Helicobacter pylori* stood out due to the high number of sequences coding for prion-like proteins (Dataset EV2). Additionally, many AD patients have experienced active and/or latent *Helicobacter* infections, pointing to a potential role in the development of the pathology (Go, 2002; Panza et al, 2019; Khaled et al, 2023). Notably, HP1 and HP2 are fragments of two putative vacuolating cytotoxins, proteins known to assemble into disrupting pores upon interaction with host cell membranes (Connolly et al, 2024). Despite their short lengths, HP1 and HP2 exhibited significant toxicity, highlighting their potential involvement in host-pathogen interactions and their contribution to disease processes.

Importantly, species encoding prion-like sequences also share phyla with others known for health benefits, such as Firmicutes (Huttenhower et al, 2012), highlighting the gut microbiota's complexity in disease development. Hence, further studies are necessary to better understand the relationship between bacterial species and neurodegenerative diseases and to explore potential therapeutic interventions targeting the gut microbiome.

The systematic platform presented here facilitated the analysis of exogenous prion-like proteins across multidisciplinary experiments, enabling the connection between structural and prionogenic properties with toxic and phenotypic effects. Our sequential engineering work corroborates the amyloid stretch hypothesis, demonstrating that the exchange of an amyloid core keeps the capacity to aggregate and propagate while modulating the prionic behavior.

Overall, while numerous protein sequences remain unidentified despite their potential health implications (Appendix Fig. S1), our findings indicate that many different sequences can influence the host and induce various effects on cognitive abilities (Fig. 6). This sequential diversity is also reflected in the increasing number of bacteria amyloid-promoting proteins with reported effects on host health(Walker et al, 2022; Wang et al, 2021; Sampson et al, 2020; Bhoite et al, 2022; Fernández-Calvet et al, 2024; Seira Curto et al, 2022). We hypothesize that, in our case, oxidative stress generation contributes to this deleterious outcome, highlighting the complex interplay between the molecules transiting the gut and neurodegenerative diseases. Our research provides valuable insights into the

role of exogenous prion-like proteins and underscores the importance of elucidating their mechanisms for identifying potential therapeutic targets and further investigating the gut-brain axis in disease pathogenesis.

# Methods

### Reagents and tools table

| Reagent/Resource | Reference or Source | Identifier or Catalog Number |
|---|---|---|
| **Experimental models** | | |
| The 74D-694 derivative yeast strain: *MATa, ade1-14^UGA, trp1-289, his3-Δ200, ura3-52, leu2-3,112 sup35::loxP* [pYK810] [PIN⁺] | Gasset-Rosa and Giraldo (2015) | NA |
| SH-SY5Y | ATCC | CRL-2266 |
| *C. elegans* N2 | Caenorhabditis Genetics Center | N2 |
| *C. elegans* CL2355 Alzheimer's model | Caenorhabditis Genetics Center | CL2355 |
| VS39 *E. coli* strain from C-DAG system | Sivanathan and Hochschild (2013) | NA |
| **Recombinant DNA** | | |
| pUKC1620 | Parham et al (2001), Gasset-Rosa and Giraldo (2015) | NA |
| pESC-URA | Agilent Technologies | 217454 |
| pVS72 | Sivanathan and Hochschild (2013) | NA |
| pVS105 | Sivanathan and Hochschild (2013) | NA |
| pet11-BH4-6HIS | GenScript | NA |
| **Oligonucleotides and sequence-based reagents** | | |
| PCR primers | This study | Table EV5 |
| **Antibodies** | | |
| Anti-GFP, rabit serum | Invitrogen™, ThermoFicher Scientific | A6455 |
| Anti-His rabbit | Thermo | RM 146 |
| Anti-rabbit horseradish peroxidase conjugate | BioRad | 1662408EDU |
| ECL KIT | Thermo | 32106 |
| **Chemicals, Enzymes and other reagents** | | |
| KOD Hot start polymerase | Novagen, Toyobo | 71086 |
| FastDigest DpnI | ThermoScientific | FD1704 |
| DMEM-F12 Glutamax | Gibco | 10565018 |
| Pen-strep | Gibco | 15140148 |
| FBS | Gibco | A5256701 |
| SH-SY5Y | ATCC | CRL-266 |
| Trypsin | Gibco | 25300-120 |
| AB40 | Genscript | RP10004 |
| Non-binding 96-well Plate | Greiner | 655906 |

| Reagent/Resource | Reference or Source | Identifier or Catalog Number |
|---|---|---|
| Thioflavin T | Sigma | 596200 |
| MTT | Fisher | 10133722 |
| DCFDA/H2DCFA | Thermo | D399 |
| DMSO | Fisher | BP231-1 |
| Congo Red | Thermo | B24310.14 |
| pET21b | Novagen | 69741 |
| HFIP | Thermo | A12747.22 |
| His Trap FF | Cytiva | 17524802 |
| **Software** | | |
| Graphpad Prism | www.graphpad.com | NA |
| Opus 7.5 | Bruker | NA |
| PSORTb v3.0 | www.psort.org | NA |
| pWALTZ | https://bioinf.uab.es/pWALTZ | NA |
| PAPA | https://combi.cs.colostate.edu/supplements/papa/ | NA |
| PLAAC | http://plaac.wi.mit.edu/ | NA |
| **Other** | | |
| EVOS M5000 Imagin System | Leica Microsystems GmbH, Mannheim, Germany | NA |
| TECAN Infinite+ NANO | TECAN Trading AG, Switzerland | NA |
| JEM-1400 | JEOL, Tokyo, Japan | NA |
| Hyperion 3000 Microscope | Bruker, Germany | NA |
| Thermo Omnic 7.1 | Thermo Scientific, Inc. | NA |
| Leica MZFLIII stereomicroscope | Leica Microsystems GmbH, Mannheim, Germany | NA |
| Vertex 70 spectrometer | Bruker, Germany | NA |

## List of positive prion-like proteins

The proteins encoded in the gut microbiome's genome were sourced from the NIH Human Microbiome Project (University of Maryland, 2009; Park et al, 2019). The original list is available at: http://downloads.hmpdacc.org/data/reference_genomes/body_sites/Gastrointestinal_tract.pep.fsa. After refining the list to remove duplicate entries, we identified 457 distinct organism names and 1,468,778 protein sequences. This shorter list is available at the figshare repository (https://doi.org/10.6084/m9.figshare.25710048.v1). This compilation of proteins underwent screening to identify prion-like proteins using two distinct algorithms: PAPA (Toombs et al, 2012) and PLAAC (Lancaster et al, 2014). These approaches seek proteins featuring prion-like domains and assign a score to each, indicating the probability of behaving as a prion.

Based on prior studies and our expertise (Iglesias et al, 2015), we identified potential prion-like proteins by employing a combination of criteria from the three different algorithms. Specifically, we employed a PAPA score threshold of 0.05 to denote a positive prion aggregation propensity. Additionally, we considered a PLAAC PRD score above 0 as indicative of sequential domains capable of initiating the aggregation process.

## Selection of amyloid core sequences

We selected 10 amyloid core segments for experimental validation. We first arranged all the sequences by their pWALTZ score to prioritize candidates with a higher propensity to self-assemble. The pWALTZ algorithm was applied to the 80-amino-acid regions previously predicted by PAPA (Iglesias et al, 2015), providing the 21-amino-acid segment with the highest aggregation propensity within it (Gil-Garcia et al, 2021; Iglesias et al, 2015; Sabate et al, 2015) (Table EV1). The candidates were chosen from those with a Q/N proportion of at least 20% and a score above 73.55, the default cut-off, to guarantee amyloid-forming sequences, both in vitro and in vivo (Sabate et al, 2015).

We excluded sequences labeled as hypothetical, those containing cysteine residues, and those with multiple transmembrane domains. We chose sequences from bacterial strains associated with genera previously reported to be altered in AD patients (Vogt et al, 2017; Panza et al, 2019; Khaled et al, 2023) and with a high likelihood of being extracellular (PSORTb tool (Tenaillon et al, 2010)).

From this shorter list of sequences that fulfill these criteria, we prioritized sequences covering the protein functions that are more enriched in prion-like proteins such as protease (RI5 and RI6 are trypsin-like), cytotoxin (HP1 and HP2 are putative vacuolating cytotoxins), DnaJ-containing domain (such as CC3 and BH4), and nucleotide-associated proteins (P9).

From the list of other functions, we chose sequences with descriptions with host interacting implications such as "penicillin-binding protein" (SA7) or that indicate aggregation such as "aggregation promoting" (LD8) or "curli associated" (HA10). When several options were left, we chose those with higher pWALTZ scores or two candidates with sequential differences (e.g., composition or length).

Overall, we constructed a diverse collection of sequences from both Gram groups, different genera, and annotated descriptions (Fig. 1D, Table EV1, Figs. EV1–2, Dataset EV3).

## Protein alignment and consensus sequence

The alignments between the 10 different amyloid-forming core candidates and the consensus sequence were obtained using the Unipro UGENE software v43.0 (Okonechnikov et al, 2012) for sequence visualization.

## Protein location prediction using PSORTb

For protein location prediction, we employed PSORTb v3.0 (Yu et al, 2010). This allowed us to anticipate the potential locations of the protein candidates. The algorithm was executed with 'bacteria' selected as the organism type, and the appropriate Gram stain was chosen for each organism. The best localization scores were used to generate Dataset EV3.

## Control peptides design

We designed eight control sequences to rule out the possibility of artificial effects during the experiments performed.

We designed the control sequences using two distinct strategies. The first strategy utilized amyloid cores with lower pWALTZ scores, while the second used the amino acid composition of proteins curated in Swiss-Prot.

For the first approach, we start by randomizing sequences with lower pWALTZ scores within the positive set. To ensure low amyloidogenic and prion-forming propensity, we implemented the following modifications:

- Substituting hydrophobic residues with glycine, an amino acid with very low prion propensity (Sabate et al, 2015).
- Maintaining a Q/N content between 9.5% and 38% while ensuring a net charge below 3, to avoid repulsive forces that might artificially inhibit aggregation and to preserve some sequence similarity with the prion-positive set.

The resulting sequences were as follows:

```
>L1
GYHEGQGGYHDGGQHGGYHGG

>L2
SGHGSHESGGSQQSRGSGGTS

>L3
NSMERNSSNSSGHGNNQDSNN

>L4
KSDQQGSHEQGMQSGMGDGMT

>L5
NNGESGGNNSGGSNNDNTSSQ
```

We developed a second set of control sequences using a Python script (see below) to generate random sequences of 21 residues with an amino acid composition similar to natural proteins based on Swiss-Prot database (https://web.expasy.org/docs/relnotes/relstat.html).

The sequences generated were:

```
>R1
GELAMARRGNPTLGGITRKRS

>R2
IDWRANVTDEPQVAAGAHTVE

>R3
KAEHTDTLMQEGARRTTDNQG
```

Python script used:

```python
import random
# Define the amino acids and their frequencies
amino_acids = [
    ('A', 8.25), ('C', 1.38), ('D', 5.46), ('E', 6.71),
    ('F', 3.86), ('G', 7.07), ('H', 2.27), ('I', 5.91),
    ('K', 5.80), ('L', 9.64), ('M', 2.41), ('N', 4.06),
    ('P', 4.74), ('Q', 3.93), ('R', 5.52), ('S', 6.65),
    ('T', 5.36), ('V', 6.85), ('W', 1.10), ('Y', 2.92)
]
# Create a list of amino acids according to their
frequency aa_list = []
for aa, freq in amino_acids:
aa_list.extend([aa] * int(freq * 100)) # Scale up for
better distribution
# Function to generate a random protein sequence of a
given length
def generate_protein_sequence(length):
return ''.join(random.choice(aa_list) for _ in
range(length))
# Generate a random sequence of 21 residues
sequence = generate_protein_sequence(21)
print(sequence)
```

Using these approaches, we obtained eight different sequences of 21 amino acids. These sequences showed negative results for aggregation and prion propensity when tested with various algorithms used in this study (PAPA score below 0.05, PLAAC PRD 0, pWALTZ score below 50) as well as Aggrescan (Conchillo-Solé et al, 2007). The pWALTZ values for these control peptides are no provided because they are below 50.0, which is the established threshold for distinguishing prion-like sequences (Zambrano et al, 2015). For this reason, the pWALTZ algorithm does not generates scores below this cutoff.

## Peptide preparation

All samples were prepared in low protein-binding microcentrifuge tubes (ThermoFisher Scientific, Waltham, MA, USA).

The amyloid-forming cores were acquired lyophilized from the Peptide Synthesis Facility, Department of Experimental and Health Sciences, Universitat Pompeu Fabra (UPF). Peptides were then solubilized in 1,1,1,3,3,3-hexafluoroisopropanol (HFIP) and separated into different aliquots and were dried overnight in a fume hood at room temperature. Different buffers and concentrations were tested to find the best conditions for amyloid aggregate formation. The best results were obtained by redissolving the peptides with DMSO to maintain them as monomers and diluting them in 50 mM phosphate buffer (PB) pH 7.4 at a final concentration ranging from 50 to 400 μM and left overnight for aggregation (see Dataset EV2).

Synthetic Aβ40 (DAEFRHDSGYEVHHQKLVFFAEDVGSNK-GAIIGLMVGGVV-NH2) was purchased from GenScript (Rijswijk, Netherlands). Stock solutions were prepared by dissolving 1 mg of the peptide to a final concentration of 250 μM and adding 20 mM sodium phosphate buffer, 0.04% $NH_3$, and NaOH to a final pH of 11. Then, the peptide was sonicated for 10 min (Fisherbrand Pittsburgh, PA, USA, FB15051) without sweep mode and stored at −80 °C. For toxicity and seeding experiments, the peptide was left to aggregate at room temperature for 24 h.

## Congo Red binding to protein aggregates

Congo Red (CR) binding to aggregated peptides was analyzed by acquiring the absorbance spectra in the 400–650 nm range using a

Cary 300 spectrophotometer (Varian). The assay was performed in a total volume of 150 μL (15 μL of aggregated peptide samples, 15 μL of CR 200 μM, and 120 μL of 50 mM PB pH 7.4). Before dye addition, all samples were sonicated for 5 min in the ultrasonic bath. Once the dye was added, samples were incubated at room temperature for 5 min before the measurement. Each spectrum was compared with that of CR alone (without peptide, as shown in Fig. 2). To avoid buffer interference, the absorbance spectrum of the buffer was subtracted from all measurements prior to their analysis. A result is considered positive if the maximum absorbance peak of CR shifts towards higher wavelengths.

## Thioflavin-T binding and aggregation kinetics

Thioflavin-T (ThT) was dissolved in Milli-Q water to 5 mM, filtered with a 0.2-μm filter, diluted to 0.5 mM, and stored at −20 °C. ThT fluorescence for aggregation kinetics was measured every 10 min and at the endpoint for ThT binding of the peptides using a 440-nm excitation filter and a 480-nm emission filter using bottom-optics in a plate reader (TECAN Infinite+ NANO). Samples were placed in a flat-bottom, black, non-binding 96-well plate (Greiner Bio-One). A total of 100 μL of sample was added per well. Each condition was measured in triplicate.

For the ThT binding experiments, the fluorescence of the peptides was measured after 72 h of aggregation of the peptides at a final concentration of 10 μM. Measurements were then compared to the negative control to determine the fold change in ThT fluorescence intensity.

For the seeding experiments, the Aβ40 peptide stock, initially at pH 11, was diluted to 25 μM in 20 mM sodium phosphate buffer and 100 mM NaCl, and the corresponding pre-aggregated peptide was added to 2.5 μM after 5 min of sonication. Control experiments were conducted under the same conditions, including controls without Aβ40 but with peptide and with just buffer and Th-T. All samples and controls were prepared in triplicate. The pH in the wells was measured both at the start and end of the experiment to ensure pH stability throughout the aggregation process and to confirm that there were no differences in the pH of the different conditions. ThT was added to a final concentration of 20 mM. The aggregation reaction was performed at 37 °C without agitation with four independent replicates. The $t_{1/2}$ is the time necessary under a given condition to reach 50% of the final fluorescence signal. The lag phase was calculated by considering its end as the point at which it reached 10% of the final fluorescence. For each condition, we measured four experimental replicates with three technical replicates each.

## Transmission electron microscopy

A 10 μl sample of aggregated peptide or *E. coli* was placed onto carbon-coated copper grids, incubated for 1 min, and dried with Whatman paper (Seira Curto et al, 2023). The grids were washed with distilled water, negatively stained with 2% (w/v) uranyl acetate for 1 min, and then dried. Micrographs were obtained using a JEM-1400 (JEOL, Tokyo, Japan) transmission electron microscope at an accelerating voltage of 80 keV.

## Infrared spectroscopy

SR-μFTIR of *C. elegans*, cell cultures, and the aggregated peptides alone was performed at the MIRAS beamline at ALBA synchrotron

(Catalonia, Spain) using a Hyperion 3000 Microscope that was equipped with a 36× magnification objective coupled to a Vertex 70 spectrometer (Bruker). The measuring range was 650–4000 cm⁻¹ and the spectra collection was carried out in transmission mode at 4 cm⁻¹ resolution, 10 μm × 10 μm aperture dimensions, and from 128–256 co-added scans. Zero filling was performed with fast Fourier transform (FFT), so that in the final spectra, there was one point every 2 cm⁻¹. Background spectra were collected from a clean area of the $CaF_2$ window every 15 min. A mercury–cadmium–telluride (MCT) detector was used, and the microscope and spectrometer were continuously purged with nitrogen gas.

*C. elegans* at the L3 stage were chosen because of the high absorbance at later stages. After 24 h of exposure to the different strains of *E. coli*, L3-worms were washed with MQ water, transferred to a $CaF_2$ window, and dried under a vacuum for 24 h. For data acquisition, a Hyperion 3000 microscope equipped with 36x magnification coupled to a Vertex 70 spectrometer (Brucker, Billerica, MA) purged with dry air with an MCT detector was used. The spectra were collected in transmission mode at 4 cm⁻¹ spectral resolution, 8 × 8 μm aperture dimensions. A total of 128 scans were co-added for each spectrum with a spectral range from 900 to 4000 cm⁻¹. SR-μFTIR data was analyzed using OPUS 7.5 (Bruker) and Unscrambler X 10.5 software (CAMO Software, Oslo, Norway). The spectra were acquired in 10 different worms per condition at different locations of the worm. Background spectra were collected from a clean area of each $CaF_2$ window.

## Fourier transform infrared (FTIR) spectral analysis

Spectra were acquired in two different ways: (a) at least 50 spectra on single cells were acquired for each sample in a given sample region; (b) maps with a dimension of minimum 50 × 50 μm with a step size of 6 × 6 μm. Fourier transform infrared (FTIR) spectra of single independent cells, the spectra from the different cell maps, and the independent spectra of amyloid aggregates without cells were analyzed using Thermo Omnic 7.1 (Thermo Scientific, Inc.) and Opus 7.5 (Bruker) software. Spectra exhibiting a low signal-to-noise ratio were eliminated.

For data processing, the second derivative of the spectra was calculated using a Savitsky–Golay algorithm with a nine-point filter and a polynomial order of 2 to eliminate the baseline contribution. Unscrambler X was used to perform PCA of the dataset. PCA was applied for the second derivative of the spectra. Unit vector normalization was applied after secondary derivation for PCA. Principal components (PCs) were calculated using the nonlinear iterative partial least squares (NIPALS) algorithm on mean-centered data. Since the PCA procedure allows weighting of the individual variables relative to each other, a constant value (1.00, equal weight) was assigned to all variables (the different wavenumbers in the 650–4000 cm⁻¹ region) as the recommended value. Ratios were calculated over the following peaks of interest: 1629 cm⁻¹ for amide I β-sheet structures (denoted $A_{1627}$), 1654 cm⁻¹, at which both the amyloid peptide and the cells have (in the derivative spectrum) a signal distinct from zero ($A_{1665}$), 1740 cm⁻¹ for ν(C=O) (carbonyl) ($A_{1740}$), 2925 cm⁻¹ for $CH_2$ asymmetric stretching vibrations ($A_{2925}$), and 2960 cm⁻¹ for $CH_3$ asymmetric stretching vibrations ($A_{2960}$). To calculate the amyloid peptide aggregation rate the ratio $A_{1629}/A_{1654}$ was calculated.

Origin 9.1 software was used for the ratio calculation, t-test analysis, and graphical representation.

For the *C. elegans* analysis, the spectra with a low signal-to-noise ratio or with a high presence of Mie scattering were eliminated. The spectra were then corrected using a concave rubberband baseline in the range 3100–1400 cm$^{-1}$. Ratios were then collected for the following peaks: 1630 cm$^{-1}$ for amide I β-sheet structures, 1654 cm$^{-1}$ for α-helices structures, 1740 cm$^{-1}$ for ν(C=O) (carbonyl), 2919 cm$^{-1}$ and 2960 cm$^{-1}$ for CH$_2$ and CH$_3$ asymmetric stretching vibrations, and 3012 cm$^{-1}$ for unsaturated bonds in the carbonate chain. To measure the amount of lipid oxidation, we calculated:

$$\frac{1741\,\text{cm}^{-1}(lipid\ oxidation)}{(2960\,\text{cm}^{-1} + 2919\,\text{cm}^{-1})(total\ lipid)}$$

and to measure the proportion of protein/lipid we calculated:

$$\frac{1654\,\text{cm}^{-1}}{(2960\,\text{cm}^{-1} + 2919\,\text{cm}^{-1})}$$

## Cell cytotoxicity assay

The SH-SY5Y cell line was authenticated by STR profiling and tested for mycoplasma contamination by ATCC prior to the start of the experiments. Following thawing, cells were cultured in Dulbecco's modified Eagle's medium/F-12 supplemented with Glutamax (DMEM/F-12 Glutamax), 10% (v/v) heat-inactivated fetal bovine serum, and 1% (v/v) penicillin/streptomycin. The cells were maintained at 37 °C and 5% CO$_2$ in a 75-cm$^2$ cell culture flask. Differentiation to neuronal cells was started 24 h after plating by replacement of the maintenance medium with differentiation culture medium for 7 days and refreshment every 72 h. The differentiation culture medium consisted of DMEM/F-12 Glutamax supplemented with 2.5% inactivated FBS and 10 μM retinoic acid. Differentiation was monitored microscopically by morphological assessment.

For the cytotoxicity assay, the cells were seeded in 96-well plates and treated at a density of 10$^4$ cells/well. The different peptides were added at 0.25–10 μM and, after 24 h of incubation, cell viability was detected by MTT assay. After removing the medium, 10 μL of 3-(4,5-dimethylthiazol-2-yl)-2,5-diphenyltetrazolium bromide (MTT) solution (5 mg/mL) and 100 μL of the medium were added to each well and incubated at 37 °C for 4 h. Then, 150 μL of dimethyl sulfoxide was added to each well to dissolve the formazan after discarding the supernatant. Absorbance values were quantified using a plate reader (TECAN Spark) at 580 nm. For each pH, the data are expressed as the percentage viability with respect to untreated cells. The untreated cells were grown in medium containing the same amount of buffer at the corresponding pH but without Aβ40 or the corresponding peptide.

For the ROS assay, the cells were seeded in 96-well plates and treated at a density of 10$^4$ cells/well. The different peptides were added at 10 μM and after 4 h of incubation, ROS were detected by DCFDA/H2DCFDA assay (Royall and Ischiropoulos, 1993).

## Expression and purification

A pET11 vector (Novagen) containing the full sequence of C9L6N5 (BH4) (GenScript) with a C-terminal His-tag was introduced into *E. coli* BL21. The bacteria were grown in LB at 37 °C. At an OD$_{600}$ of 0.5, 1 mM IPTG was added and then the culture was left overnight at 22 °C. Afterward, the cells were centrifugated, and the pellet was resuspended in PBS with 5 mM imidazole, RNAse A, and DNAse I and sonicated. Then the lysate was ultracentrifuged and the supernatant was added to a His-Trap FF (Cytiva, Barcelona, Spain) column, which was washed with a gradient of PBS with imidazole buffer, reaching a maximum concentration of 200 mM imidazole. The protein purity was assessed with an SDS-PAGE gel and a Western blot to ensure it was the protein of interest.

The purified samples were separated by SDS-PAGE gels in duplicate. One was stained with Coomassie blue and the other was transferred to nitrocellulose membranes. After the transfer, the membranes were blocked with 4% bovine serum albumin. The membrane was incubated with an anti-His rabbit primary antibody (Thermo Scientific; 1:1000). The secondary antibody was an anti-rabbit horseradish peroxidase conjugate (Bio-Rad; 1:3000). The reaction was developed with an ECL kit (Thermo Scientific).

## Yeast strain and nonsense suppression system

The yeast strain used in this project was derived from 74D-694 *MATa, ade1-14$^{UGA}$, trp1-289, his3Δ-200, ura3-52, leu2-3,112 sup35::loxP* [pYK810]) [PIN$^{\pm}$] (Von Der Haar et al, 2007). The N-terminal Sup35p modified chimeras (where the residues 2-40 are replaced with 21-residue peptides) were encoded in a centromeric pUCK1620 plasmid bearing a *HIS3* selection marker (Dataset EV6 and Reagents and Tools table). The strain and the original SUP35-pUCK1620 vector was kindly provided by Dr R. Giraldo (Gasset-Rosa and Giraldo, 2015).

In our assay, the full-length Sup35 version and their peptide-containing chimeras (Fig. 4A) are expressed constitutively at physiological levels to maintain essential terminator factor activity. The original full-length Sup35 was exchanged with the new peptide-containing chimeras by plasmid shuffling with 5-FOA (Fan and Xiao, 2021). To prompt aggregation, an additional copy of only the N and M domains fused to GFP was transiently overproduced under a galactose-inducible promoter in the vector pESC-URA (Agilent Technologies). This strategy allowed us to induce prion conversion when necessary and simultaneously monitor protein aggregation using fluorescent microscopy. The initial 40 residues of Sup35p and the NM domain fused to GFP were replaced with the selected amyloid cores (Fig. 3A).

In the nonsense suppression assay, the *ADE1* gene (*ade*1-14), causes the accumulation of a red pigment (an adenine precursor) that colors colonies red when Sup35 is soluble (non-prion colonies [*psi$^-$*]), and white or pink when it becomes insoluble (prion colonies [*PSI$^+$*]). Since the conversion to [*PSI$^+$*] is a rare event, we induced it by over-expressing the NM segment of Sup35 fused to GPF (Appendix Supplementary Data); this increases the rate of prion conversion and facilitates monitoring of the aggregation inside the cell.

## Fluorescent foci detection in yeast

Yeast cells grown for 3–5 days on SD medium plates were inoculated into synthetic medium with raffinose (instead of glucose) and without histidine or uracil (SRaf -His, -Ura), and subsequently grown at 30 °C under vigorous agitation for 3 days. To induce expression of NM-GFP chimeras, yeast was inoculated

into synthetic medium with 2% galactose and 2% raffinose 1 day before the GFP assay as they are under the *GAL10* promoter. On the day of the experiment, the cells were examined and recorded using fluorescence microscopy (EVOS M5000) on a 96-well plate.

## Nonsense suppression assay

In this assay, a preculture was initiated in SD -His -Ura medium, using 2% glucose as the carbon source, for 3 days at 30 °C. This 3-day interval serves to repress expression prior to the onset of the assay. Subsequently, yeast cells were washed with PBS buffer and induced to express the NM-GFP fusions by inoculating with fresh media containing 2% galactose and 2% raffinose and incubated two days at 30 °C. All cultures were adjusted to an $OD_{600}$ of 1 and plated at a 1/1000 dilution on ¼ YPD plates (1% bacto-yeast extract, 2% bacto-peptone, 2% glucose, 2% bacto-agar), then incubated for 4–5 days at 30 °C. After incubation, plates were transferred to 4 °C for an additional 5–7 days and subsequently examined for red and white colony phenotypes. Images were captured for each plate using a Leica MZFLIII stereomicroscope (Leica Microsystems GmbH, Mannheim, Germany) at 8× magnification. All assays were conducted in at least triplicate with colony counts exceeding 200 per plate. Colony counting was carried out for each assay, followed by t-student analysis for comparison against the negative control. Red colonies were indicative of the [*psi*⁻] phenotype, while white colonies indicated a stable [*PSI*⁺] phenotype characterized by efficient transmission between cells. Pink colonies were observed as indicative of an unstable [*PSI*⁺] phenotype, attributed to factors including low transmission between cells and/or high reversion between the aggregated and soluble forms of Sup35p.

## Colony-color phenotype assay to detect extracellular aggregates

For the expression and extracellular aggregation of the Sup35p chimeras, we utilized the *E. coli* strain VS39 (kindly provided by Ann Hochschild). This strain is deficient in curli genes (*csgA*, *csgB*, and *csgC*), resistant to kanamycin, and contains a pACYC-derived plasmid named pVS76. This plasmid orchestrates the synthesis of the outer-membrane curli protein, CsgG, regulated by an IPTG-inducible promoter. Additionally, VS39 carries a *cat* gene that is constitutively transcribed and provides resistance to chloramphenicol.

The *Sup35NM* genes were obtained by modifying the pEXPORT plasmid pVS72, which harbors an intact *Sup35NM*. The final plasmids were subsequently transformed into the VS39 strain. The detailed sequence information of the final constructs is provided in the Dataset EV6 and Reagents and Tools table. In this study, we also incorporated the pEXPORT plasmid pVS105 encoding the Sup35 medium (Sup35M*)* region of the gene.

To measure the capacity to form extracellular aggregates among the various Sup35p variants, bacterial cells were spotted on CR-inducing plates. These plates consist of LB agar supplemented with 100 μg/ml carbenicillin, 25 μg/ml chloramphenicol, 0.2% w/v L-arabinose, 1 mM IPTG, and 10 μg/ml CR. Following a 5-day incubation period at 22 °C, the plates were subjected to imaging to assess the extent of CR binding. To obtain quantitative data, red

color intensity values were derived from quadruplicate samples using ImageJ (Nguyen et al, 2014).

## *C. elegans* strains and maintenance

We utilized the wild-type N2 strain (dvIs50 [pCL45 (snb-1::Abeta 1-42::3′ UTR(long) + mtl-2::GFP]) and the strain CL2355 expressing Aβ42 pan-neuronal (dvIs50 [pCL45 (snb-1::Abeta 1-42::3′ UTR(long) + mtl-2::GFP]) both obtained from the Caenorhabditis Genetics Center (CGC). During the experiments, the nematodes were incubated at 20 °C. For control conditions, the worms were grown on standard nematode growth medium (NGM) plates that had been seeded with *E. coli* OP50 as a food source. When the C-DAG *E. coli* strains were used, NGM plates were prepared with antibiotics (Carbenicillin and Chloramphenicol), and inductors (Arabinose and IPTG). These plates were seeded with bacteria cultures expressing the C-DAG system to enhance the presence of extracellular aggregates.

To synchronize the nematode population, we subjected them to 5% hypochlorite treatment, followed by a 24-h rotation at 80 rpm in M9 buffer without any food source. Following this procedure and until they reached the young adult stage, each group of worms was incubated with the respective bacterial strain.

## *C. elegans* butanone associative short-term learning assay (STAM)

The STAM employed three replicates of a starting population of starved worms that were subsequently divided and subjected to three distinct conditions, with 300–500 worms each, prior to the chemotaxis assay (Fig. 6A). These conditions were (i) a naive population that had not been conditioned with the odorant, (ii) a population tested immediately after conditioning (learning index), and (iii) a population tested an hour after conditioning (memory index). The chemotaxis assay was conducted on plates on which each differently treated population was positioned equidistantly from a spot containing $NaN_3$ and butanone (the chemoattractant) and another spot containing $NaN_3$ and ethanol, a compound to which the worms had not been previously exposed (Stein and Murphy, 2014; Kauffman et al, 2011).

The butanone associative learning assay was performed following established methods (Kauffman et al, 2011). Young adults were washed in M9 and collected by gravity sedimentation in a 15-mL conical tube to remove bacteria. Some animals were immediately transferred to the chemotaxis plate to assay the naive condition (CI (naive)). The rest were starved in M9 for 1 h. After the starvation, animals were conditioned for 1 h on NGM plates seeded with the corresponding *E. coli* strain and containing a 2-μL drop of 10% butanone, diluted in absolute ethanol, on the lid.

The conditioned worms (immediately after conditioning, learning index (LI), or after 1 h of starvation, memory index (MI)) were washed with M9 and transferred to the chemotaxis plate. For the chemotaxis assay, 10 cm unseeded NGM plates were used, and three circles (each 1 cm in diameter) on the bottom and both sides of the plate were marked. Then, 1 μL of 1 M sodium azide and 1 μL of 10% of butanone were added to the left spot, and the same amount of sodium azide and pure ethanol control were added to the right spot. The animals were placed on the bottom

spot. After 1 h, the numbers of worms located in the butanone and ethanol spots as well as at the original bottom spots were counted. Each chemotaxis assay was performed with 300–500 worms and in triplicate.

Our standard formula to calculate the chemotaxis index is as follows:

CI = [(Number of worms in the butanone zone) – (Number of worms in the ethanol zone)]/(Total number of worms)

With unconditioned worms, we calculated the CI (naive):

CI (naive) = [(Number of worms in the butanone zone) – (Number of worms in the ethanol zone)]/(Total number of worms)

Immediately after conditioning, we calculated the learning index as:

LI = CI (right after conditioning) – CI (naive)

After conditioning followed by 1 h of starvation, we calculated the memory index as:

MI = CI (after conditioning and starvation) – CI (naive).

### *C. elegans* gut granule autofluorescence detection

The EVOS microscope (EVOSTM M5000 Imaging System) was employed to capture autofluorescence images of *C. elegans* using a green fluorescent protein (GFP) filter and a transmission filter. The images were acquired from animals treated identically to those used in the chemotaxis assays. To visualize gut granules, we centered the images on the first two intestine rings, which correspond to the region with the highest fluorescence intensity. Subsequently, ImageJ software was used for image analysis. The selected region for analysis was carefully chosen to avoid the presence of the gonads, thus mitigating potential interference.

### Statistics

GraphPad PRISM 8 software was used for statistical analysis of the obtained data.

### Blinding statement

Blinding procedures were partially implemented in this study. While the experimental design and peptide selection were based on computational aggregation propensity predictions (pWALTZ), the researchers conducting the experimental assays (including ThT fluorescence, TEM imaging, toxicity assays, and behavioral assays in *C. elegans*) were provided only with sample identifiers (peptide or strain names) without prior knowledge of their predicted aggregation propensities. This ensured that data collection and analysis were conducted without bias regarding the expected amyloid-forming potential of each sequence. Full double-blinding was not possible due to the necessity of experimental design oversight.

## Data availability

The datasets used and generated in this study are available in the following databases: Gut microbiome sequences (filtered single-domain list): Figshare (https://doi.org/10.6084/m9.figshare.25710048.v1).

The source data of this paper are collected in the following database record: biostudies:S-SCDT-10_1038-S44320-025-00114-4.

## Peer review information

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

## Acknowledgements

This work was supported by: the grant FPU21/03897 funded by MICIU/AEI/ 10.13039/501100011033 and by FSE+; the grant RYC2019-026752-I funded by MCIN/AEI/10.13039/501100011033 and by FSE invierte en tu futuro; the project PID2020-117454RA-I00 funded by MCIN/ AEI/10.13039/ 501100011033; the project CNS2023-144437 funded by MICIU/AEI/ 10.13039/501100011033 and by the European Union – NextGenerationEU/ PRTR; and by L'Oréal-UNESCO For Women in Science Programme. We extend our gratitude to the following individuals: Ann Hochschild and Padraig Deighan for kindly providing the SV39 *E. coli* strain and plasmids pVS105 and PVS72. Rafael Giraldo for generously providing the 74D-694 S. cerevisiae strain. Antonela Lavatelli for her assistance in resolving basic experimental issues. Julian Cerón and Antonio Miranda-Vizuete for their invaluable responses to our inquiries. Anna Laromaine for sharing her *C. elegans* laboratory expertise and aiding in the construction of the initial picker. Esther Dalfo for enlightening us about the requirements of a *C. elegans* laboratory and for sharing valuable information regarding contacts and databases related to *C. elegans*. Gian G. Tartaglia for his inestimable support to start the project. Nuria Benseny for her assistance in the microFTIR experiments. Josep Cladera for his responses and discussions around the investigation. *C. elegans* N2, CL2355 and *E. coli* OP50 were provided by the CGC, which is funded by NIH Office of Research Infrastructure Programs (P40 OD010440). µFTIR experiments were performed at MIRAS beamline at ALBA Synchrotron with the collaboration of ALBA staff. Transmission electron microscopy (TEM) images were acquired at the Servei de Microscòpia i Difracció de Raigs X (SMiDRX) of the Universitat Autònoma de Barcelona (UAB).

## Author contributions

**Jofre Seira Curto**: Data curation; Formal analysis; Validation; Investigation; Methodology; Writing—original draft; Writing—review and editing. **Adan Dominguez Martinez**:Formal analysis; Supervision; Investigation; Methodology. **Genis Perez Collell**: Data curation; Investigation; Methodology. **Estrella Barniol Simon**: Investigation. **Marina Romero Ruiz**: Investigation. **Berta Franco Bordés**: Investigation. **Paula Sotillo Sotillo**: Investigation; Methodology. **Sandra Villegas Hernandez**: Conceptualization; Writing—review and editing. **Maria Rosario Fernandez**: Conceptualization; Resources; Data curation; Formal analysis; Supervision; Validation; Investigation; Methodology; Writing—original draft; Writing—review and editing. **Natalia Sanchez de Groot**: Conceptualization; Resources; Data curation; Formal analysis; Supervision; Funding acquisition; Validation; Writing—original draft; Project administration; Writing—review and editing.

Source data underlying figure panels in this paper may have individual authorship assigned. Where available, figure panel/source data authorship is listed in the following database record: biostudies:S-SCDT-10_1038-S44320-025-00114-4.

## Disclosure and competing interests statement

The authors declare no competing interests.

# Expanded View Figures

**Figure EV1.  Sequential information regarding the ten prion-like sequences selected from bacteria found in the gut.**

Diagram showing the prion-like regions predicted for all the amyloid cores selected. In green are the main sequences. In light blue are the prion domains (PrD) predicted by PLAAC. In dark blue are the prion domains predicted by PAPA. In purple, the amyloid cores predicted by pWALTZ. Turquoise and yellow are other domains.

## M5YJZ4-HP1

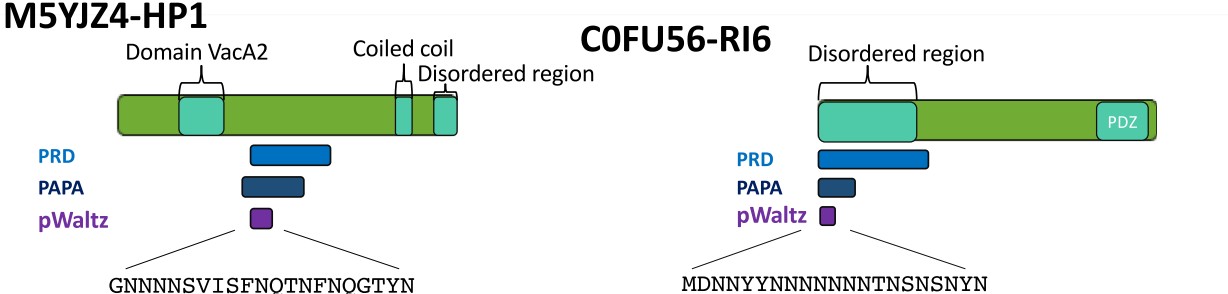

GNNNNSVISFNQTNFNQGTYN

## M3SJI9-HP2

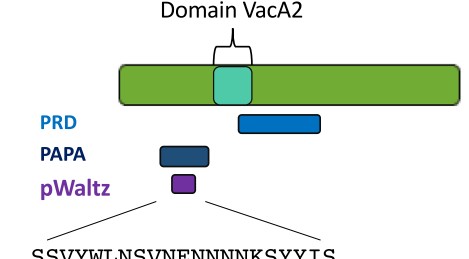

SSVYWLNSVNENNNNKSYYIS

## C0B555-CC3

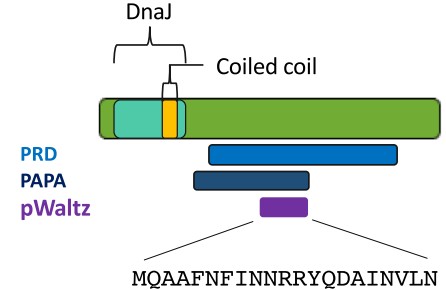

MQAAFNFINNRRYQDAINVLN

## C9L6N5-BH4

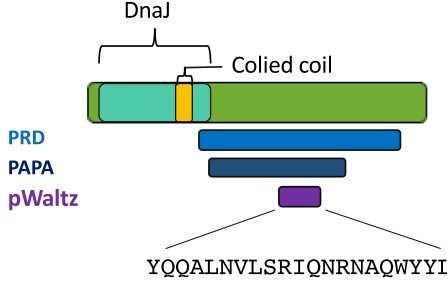

YQQALNVLSRIQNRNAQWYYL

## D4KZ46-RI5

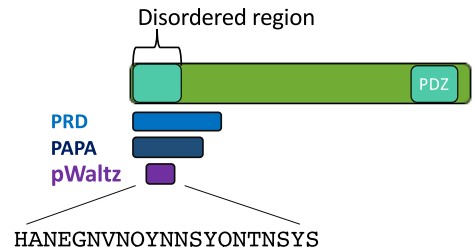

HANEGNVNQYNNSYQNTNSYS

## C0FU56-RI6

MDNNYYNNNNNNNTNSNSNYN

## E7GXS5-SA7

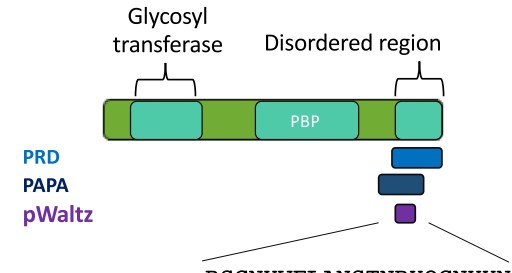

RSGNYVFLANSTNRYQGNYYN

## F0HUU2-LD8

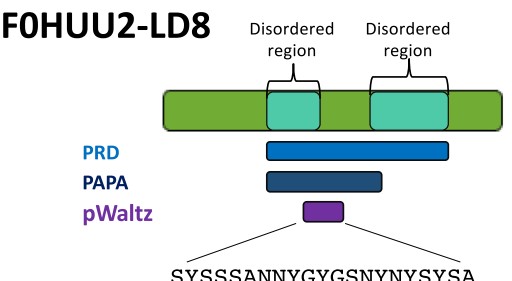

SYSSSANNYGYGSNYNYSYSA

## F5LFB6-P9

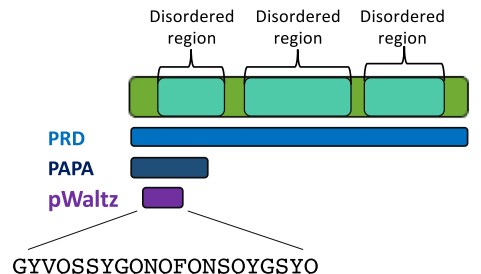

GYVQSSYGQNQFQNSQYGSYQ

## G9Y7N7-HA10

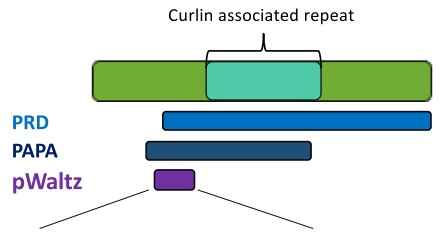

STGYSDLNSTYNQSALINQIG

## M5YJZ4-HP1

## C0FU56-RI6

## M3SJI9-HP2

## E7GXS5-SA7

## C0B555-CC3

## F0HUU2-LD8

## C9L6N5-BH4

## F5LFB6-P9

## D4KZ46-5RI

## G9Y7N7-HA10

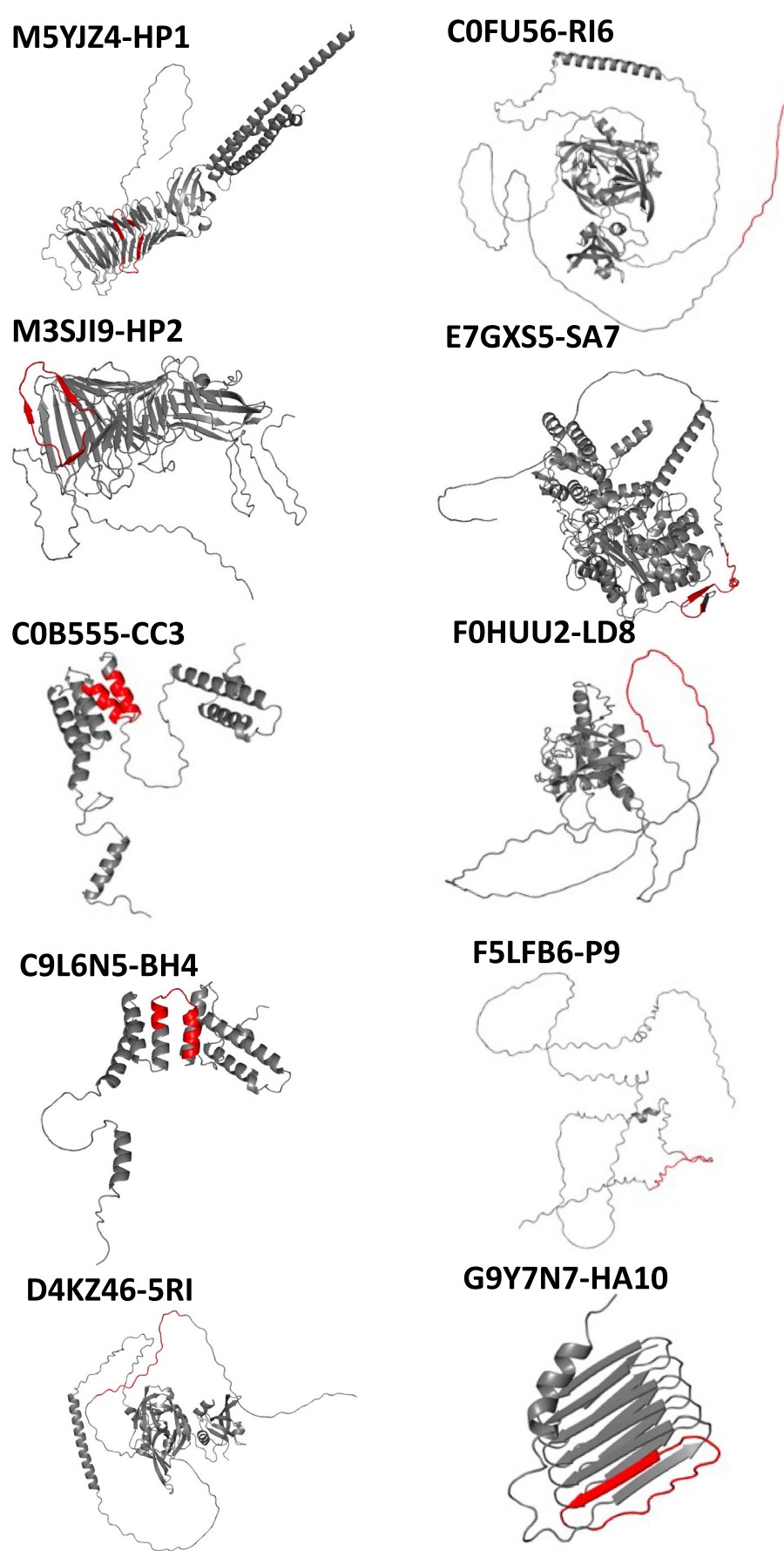

**Figure EV2.   Predicted structure of the ten prion-like sequences selected from bacteria found in the gut.**

The images show the Alpha-fold prediction of the whole protein structure.

