## [Peer Review File · Molecular Systems Biology]

Exogenous Prion-Like Proteins and Their Potential to Trigger Cognitive Dysfunction

Jofre Seira Curto, Adan Dominguez Martinez, Genis Perez Collell, Estrella Barniol Simon, Marina Romero Ruiz, Berta Franco Bordés, Paula Sotillo Sotillo, Sandra Villegas Hernandez, María Fernandez, and Natalia Sánchez de Groot

Corresponding authors: Natalia Sánchez de Groot (natalia.sanchez@uab.cat) , María Fernandez (rosario.fernandez@uab.cat)

Review Timeline:

Submission Date:	3rd May 24
Editorial Decision:	28th Jun 24
Revision Received:	4th Jan 25
Editorial Decision:	11th Mar 25
Revision Received:	16th Apr 25
Accepted:	2nd May 25

Editor: Poonam Bheda

Transaction Report:

28th Jun 2024

Manuscript Number: MSB-2024-12399

Title: Exogenous Prion-Like Proteins and Their Potential to Trigger Cognitive Dysfunction

Dear Dr Sánchez de Groot,

Thank you again for submitting your work to Molecular Systems Biology. We have now heard back from the three reviewers who agreed to evaluate your study. As you will see below, the reviewers appreciate that the proposed approach addresses a timely topic. However, they raise a series of concerns, which we would ask you to address in a major revision.

Without repeating all the comments listed below, some of the more fundamental issues raised are the following:

- Appropriate negative and positive controls as suggested by Reviewers 2 and 3 should be included
- Experiments should be performed with biological replicates and clearly indicated in the manuscript

Editorially, however, we would be willing to overrule Reviewer 1 to leave the *C. elegans* work in the manuscript and we would also overrule Reviewer 2 who has suggested extending the work to vertebrates.

All other issues raised would need to be satisfactorily addressed. Please let me know in case you would like to discuss in further detail any of the comments, I would be happy to schedule a call.

We require:

1) A .docx formatted version of the manuscript text (including legends for main figures, EV figures and tables). Please make sure that the changes are highlighted to be clearly visible. Alternatively you may choose to submit your manuscript as a LaTeX file.

4) A .docx formatted letter INCLUDING the reviewers' reports and your detailed point-by-point responses to their comments. As part of the EMBO Press transparent editorial process, the point-by-point response is part of the Peer Review File (PRF), which will be published alongside your paper.

5) A complete author checklist, which you can download from our author guidelines (<https://www.embopress.org/page/journal/17574684/authorguide#submissionofrevisions>). Please insert information in the checklist that is also reflected in the manuscript. The completed author checklist will also be part of the PRF.

6) Please note that all corresponding authors are required to supply an ORCID ID for their name upon submission of a revised manuscript.

7) It is mandatory to include a 'Data Availability' section after the Materials and Methods. Before submitting your revision, primary datasets produced in this study need to be deposited in an appropriate public database, and the accession numbers and database listed under 'Data Availability'. Please remember to provide a reviewer password if the datasets are not yet public (see <https://www.embopress.org/page/journal/17574684/authorguide#dataavailability>).

This study includes no data deposited in external repositories.

8) All Materials and Methods need to be described in the main text using our 'Structured Methods' format, which is required for all research articles. According to this format, the Methods section includes a Reagents and Tools Table (listing key reagents, experimental models, software and relevant equipment and including their sources and relevant identifiers) followed by a Methods and Protocols section describing the methods using a step-by-step protocol format. The aim is to facilitate adoption of the methodologies across labs. More information on how to adhere to this format as well as a downloadable template (.docx) for the Reagents and Tools Table can be found in our author guidelines:

<https://www.embopress.org/page/journal/17444292/authorguide#structuredmethods>

9) For data quantification: please specify the name of the statistical test used to generate error bars and P values, the number (n) of independent experiments (specify technical or biological replicates) underlying each data point and the test used to calculate p-values in each figure legend. The figure legends should contain a basic description of n, P and the test applied. Graphs must include a description of the bars and the error bars (s.d., s.e.m.). Please provide exact p values.

10) Our journal encourages inclusion of *data citations in the reference list* to directly cite datasets that were re-used and obtained from public databases. Data citations in the article text are distinct from normal bibliographical citations and should directly link to the database records from which the data can be accessed. In the main text, data citations are formatted as follows: "Data ref: Smith et al, 2001" or "Data ref: NCBI Sequence Read Archive PRJNA342805, 2017". In the Reference list, data citations must be labeled with "[DATASET]". A data reference must provide the database name, accession number/identifiers and a resolvable link to the landing page from which the data can be accessed at the end of the reference. Further instructions are available at .

11) We replaced Supplementary Information with Expanded View (EV) Figures and Tables that are collapsible/expandable online. A maximum of 5 EV Figures can be typeset. EV Figures should be cited as 'Figure EV1, Figure EV2' etc... in the text and their respective legends should be included in the main text after the legends of regular figures.

<https://www.embopress.org/page/journal/17574684/authorguide#expandedview>

13) Author contributions: CRediT has replaced the traditional author contributions section because it offers a systematic machine readable author contributions format that allows for more effective research assessment. Please remove the Authors Contributions from the manuscript and use the free text boxes beneath each contributing author's name in our system to add specific details on the author's contribution. More information is available in our guide to authors.

14) Disclosure statement and competing interests: We updated our journal's competing interests policy in January 2022 and request authors to consider both actual and perceived competing interests. Please review the policy

<https://www.embopress.org/competing-interests> and update your competing interests if necessary.

Share synopsis text and image, as well as eTOC:

Please note that these would be the final versions and changes during proofing are usually not allowed

16) As part of the EMBO Publications transparent editorial process initiative (see our policy here:

https://www.embopress.org/transparent-process#Review_Process), Molecular Systems Biology will publish online a Peer Review File (PRF) to accompany accepted manuscripts.

In the event of acceptance, this file will be published in conjunction with your paper and will include the anonymous referee reports, your point-by-point response and all pertinent correspondence relating to the manuscript. Let us know whether you agree with the publication of the PRF and as here, if you want to remove or not any figures from it prior to publication.

Please note that the Authors checklist will be published at the end of the PRF.

Molecular Systems Biology has a "scooping protection" policy, whereby similar findings that are published by others during review or revision are not a criterion for rejection. Should you decide to submit a revised version, I do ask that you get in touch after three months if you have not completed it, to update us on the status.

Yours sincerely,

Poonam Bheda, PhD
Scientific Editor
Molecular Systems Biology

Reviewer #1:

The authors aim at studying the interaction of pathogenic human prionoids and non-mammalian gut prions using i.a. modified Sup35 to explore relevance of the brain-gut axis for prion-interactions. They are using multiple approaches for this endeavor, including a *C. elegans* model.

The first identify amyloid aggregation sequences from gut bacteria using a computer model to next test their co-aggregation with A β in vitro. They then study the toxicity of these prion-like sequences in cell cultures.

They then evaluate the aggregation propensity of the identified aggregation prone sequences by expressing them as replacement for the amyloid-sequence of Sup35, using Sup35NM and delta-Sup35 as positive/negative controls.

Eventually, they study the effect of the identified prion-sequences in a *C. elegans* model.

In summary, they are able to show that potentially prionogenic sequences present in the gut microbiome, e.g. such sequences derived from *Helicobacter pylori*, are able to provoke prion aggregation and cause neurotoxic effects in vitro and in *C. elegans*. The claim that lipid oxidation is playing a role in the pathomechanism in *C. elegans*.

The study benefits from a complex approach to an interesting question.

Major comments:

- 1) The models used are all rather artificial. The prion-sequences used for the study are based on calculations, not on naturally occurring/known prions. This limitation should be discussed.
- 2) In contrast to the artificial models used here, trans-seeding of the naturally occurring Sup35 prion with Alzheimer's tau protein has recently been shown in vivo. This study should be cited (Flach et al. AD 2022)
- 3) The key ideas of the study, including the identification of theoretically prionogenic sequences in the microbiome, their link to human diseases e.g. *Helicobacter pylori* infections (a frequent and relevant disease) and their toxicity remain difficult to be found in the exhausting lengths and wide complexity of the paper. Although the plethora of approaches and ideas is impressive, reducing the data set to the most relevant sequences and providing the other data in Suppl Data would help the understanding.
- 4) The connection of the observed effects in *C. elegans* to the amyloid-toxicity are difficult to understand and the proposed mechanism (lipid oxidation) does not convince me. I could easily live without the *C. elegans* part of the manuscript. The manuscript should then focus on the Amyloid-interaction of the identified sequences.

Minor Comments:

- 1) Figure 2A : Some TEM figures hardly allow to recognize fibrils.
- 2) Figure 3A : The differences in lag-times seem minor. The assay lacks a negative (inhibitory) control.
- 3) The identified sequences should be labelled according to their "logical" origin e.g. *Helicobacter pylori* "HP1" etc. *Coprococcus* comes "CC1"....
- 4) The abstract should focus on the relevant logic of the manuscript, e.g. the identification of *Helicobacter pylori* sequences, their co-aggregation effects with A β , and their effect on neuronal cells, including the limitation of the study, by using artificial sequence fragments without the presence of corresponding proteins in nature.

Reviewer #2:

In this study, Curto et al. investigated the biophysical and pathogenesis properties of ten gut microbe prion-like amyloid-forming core sequences. First, the authors identified prion-like sequences from gut bacteria through bioinformatic analysis. Then, they analyzed the aggregation properties using transmission electron microscopy, Fourier-transform infrared, spectroscopy, and binding to the amyloid dyes thioflavin-T and Congo red, and found that these core peptide sequences can self-assemble to amyloid fibrils and promote soluble A β 40 aggregation in vitro. Finally, Curto et al. tested the potential pathological role in cell culture, yeast and *C. elegans* and found that all peptides tested promote aggregation and display toxic effects. The finding of pathogenic roles of exogenous amyloid-forming core peptides is really interesting and consistent with Wang's reports regarding the effect of bacterial curli amyloid fibril on *C. elegans* published in 2021 (PNAS, 118(34)). However, the data is preliminary.

Some experiments require better controls and the mechanisms underlying the toxicity of these peptides have not been addressed. And it would be more interesting to know if the same effects exist in vertebrates.

Major points

1. In this study, the authors analyzed the aggregation and pathogenic function of 10 amyloid-forming core sequences. However, no negative control is included. Therefore, it is hard to exclude the possibility of artificial effects. I strongly recommend the authors add several negative controls (around 5 random sequences with the same length as the amyloid-forming core sequence).
2. Fig. 2, what is the concentration for the experiment? Have the experiments been replicated (for B and D)? If not, please do repeat.
3. Fig. 3, In cell culture, the aggregation properties of the prion-like core peptides are not correlated with the ROS or cytotoxicity. What is the role of the aggregation in the pathogenesis?
4. Fig. 4, all chimeric Sup35s aggregate, which can't exclude the possibility that a random sequence in the N-terminal could induce the aggregation. A negative control is required to exclude the possibility.
5. Only one full-length protein (C9L6N5, a protein with a DNAJ domain) was tested for the role in A β 40 aggregation, which is not typical of ThT fluorescence.
6. Statistics are recommended for Figure 2C, Figure 3E, Figure 4D, Figure 6D-E, and Fig 7B-C. And correlation analysis between oxidation, gut granules and memory would be helpful.

Minor points

There is a mistake in "These core sequences are rich in amino acids that despite being associated with disordered conformations also keep amyloid propensities such as asparagine (N), glutamine (N), and tyrosine (Y)". The abbreviation for glutamine is Q.

Reviewer #3:

The findings by S. Curto et al. suggest that ingested or exogenous prion-like protein derived from gut bacteria can form amyloids, which interfere with the aggregation of amyloid beta peptide, resulting in loss of sensory memory and increased lipid oxidation. These findings are interesting and thought-provoking as they highlight a potential link between gut microbiota as a reservoir of exogenous prion-like sequences and cognitive dysfunction. The experiments are well-planned and executed with proper controls and replicates. Their findings could be useful in other neurodegenerative diseases where the gut microbiome may play an important role. I have listed a few major and minor comments that may help to improve the current version of the manuscript.

Major comments:

1. The authors have screened prion-like sequences using the following algorithms: PAPA, PLAAC, and pWALTZ in the gut microbiome and identify disordered sequences with a prion-like composition and provide a score that reflects the probability of a sequence behaving as a prion. I wondered if the authors compared the sequence with already reported CsgA-like proteins from the gut, as mentioned in Bhoite et al. *J Biol Chem.* 2022 (10.1016/j.jbc.2022.102088). It would be great to see any sequence similarity between the peptides identified in this work and the proteins reported by Bhoite et. al.
2. Authors have shown a robust correlation between peptide net charge and lag time kinetics. I am curious if authors tried to understand the nature of the interaction between peptide and A β (like electrostatic interaction, which can be done in the presence of NaCl) as authors have shown more positive charge results in faster A β 40 aggregation and more negative charge results in slower aggregation.
3. Authors have shown cellular ROS production in which six out of ten peptides produced ROS compared to control without aggregates. However, it might be useful to add a positive control.
4. C1 and C2 exhibited higher toxicity, which are fragments of two putative vacuolating cytotoxins and assembled into oligomers upon interaction with cellular membrane. But, somehow, it is not very clear to me how the authors come to this conclusion of interaction. It might be a good idea to add this to the discussion.
5. The main forms of A β in amyloid deposits are 40 and 42. Authors have performed experiments on prion-like peptides with A β 40. It would be great if the authors could either add few experiments with A β 42 or provide their speculation in the discussion.

Minor comments:

1. Table 1: Last line, s is missing in species.
2. In the result section Ingestion of exogenous prion-like proteins increases lipid oxidation - 2nd line from the last "with" is written two times.
3. 3rd line in the 3rd paragraph (Without ab40 and with peptide and without ab 40 or peptide) is repeated in the Thioflavin T binding and aggregation kinetics section of material and methods.
4. In nonsense suppression assay of material and method section, 2 is written in both numeric and word in the sentence.
5. Figure 2: Absorvance to be replaced by Absorbance, what is the control in CR plot?
6. Bacterial species are to be written in italics throughout the text.

Reviewer #1:

The authors aim at studying the interaction of pathogenic human prionoids and non-mammalian gut prions using i.a. modified Sup35 to explore relevance of the brain-gut axis for prion-interactions. They are using multiple approaches for this endeavor, including a *C. elegans* model.

The first identify amyloid aggregation sequences from gut bacteria using a computer model to next test their co-aggregation with Abeta in vitro. They then study the toxicity of these prion-like sequences in cell cultures.

They then evaluate the aggregation propensity of the identified aggregation prone sequences by expressing them as replacement for the amyloid-sequence of Sup35, using Sup35NM and delta-Sup35 as positive/negative controls.

Eventually, they study the effect of the identified prion-sequences in a *C. elegans* model.

In summary, they are able to show that potentially prionogenic sequences present in the gut microbiome, e.g. such sequences derived from *Helicobacter pylori*, are able to provoke prion aggregation and cause neurotoxic effects in vitro and in *C. elegans*. The claim that lipid oxidation is playing a role in the pathomechanism in *C. elegans*.

The study benefits from a complex approach to an interesting question.

We sincerely thank the reviewer for their thoughtful and positive assessment of our work. We appreciate their recognition of our multidisciplinary approach and the relevance of our findings to the brain-gut axis and prion interactions. Their encouraging comments about the study's complexity and model integration are greatly valued. We have carefully addressed all suggestions and hope the revisions enhance the clarity and impact of the manuscript.

1) The models used are all rather artificial. The prion-sequences used for the study are based on calculations, not on naturally occurring/known prions. This limitation should be discussed.

We agree with the reviewer that it is important to address the limitations associated with the synthetic model used in this study. To address this, we have modified the text in the manuscript to explicitly acknowledge the artificial nature of the models and discuss their implications.

In the Abstract:

*"The gut is exposed to a wide range of proteins, including ingested proteins and those produced by the resident microbiota. While ingested prion-like proteins can propagate across species, their implications for disease development remain largely unknown. Here, we apply a multidisciplinary approach to examine the relationship between the biophysical properties of exogenous prion-like proteins and the phenotypic consequences of ingesting them. **Through computational analysis** of gut bacterial proteins, we identified an enrichment of prion-like sequences in *Helicobacter pylori*. Based on these findings, **we rationally designed a set of synthetic prion-like sequences** that form amyloid fibrils, interfere with amyloid-beta-peptide aggregation, and trigger prion propagation when introduced in the yeast Sup35 model. When *C. elegans* were fed a bacterial expressing these prion-like proteins, they lost sensory memory and increased lipid oxidation. These data suggest a link between memory impairment, the conformational state of aggregates, and oxidative stress. Overall, this work supports gut microbiota as a reservoir of exogenous prion-like sequences, especially *H. pylori*, and the gut as an entry point for molecules capable of triggering cognitive dysfunction."*

In the **Introduction**:

*“Our **computational** work detected sequences with prion-like properties in 63% of the species classified as from the GI tract in an NIH Human Microbiome Project (NIH-HMP) dataset, with a special enrichment of prion-like sequences in *Helicobacter pylori* (REF). The amyloid cores of these sequences were rationally applied in a series of interconnected in vitro, and in vivo analyses. We **designed a set of synthetic prion-like proteins** to be introduced into the *C. elegans* digestive tract, which led to sensory memory impairment and lipid oxidation.”*

In the **Discussion**:

*“To explore the impact of exogenous prion-like proteins on the host, we assembled a collection of potential amyloidogenic sequences. **These sequences were computationally designed based on gut microbiome analysis, providing a synthetic and simplified approach that does not intend to represent any specific natural protein** but instead aims to offer a broad overview of prion-like sequences.”*

2) In contrast to the artificial models used here, trans-seeding of the naturally occurring Sup35 prion with Alzheimer's tau protein has recently been shown in vivo. This study should be cited (Flach et al. AD 2022).

We thank the reviewer for this comment. We agree that highlighting the relevance of the Sup35 prion in human disease models is important. Accordingly, we have cited Flach et al. (AD 2022) together with Meng et al. (Sci. Adv. 2023) in the text, which have been added to the “Bacterial amyloid cores from the gut are functional in yeast” section:

*“The exchange of prion-like domains is commonly studied in the yeast prion Sup35 since it allows monitoring protein aggregation and propagation easily through an in vivo nonsense suppression assay. Additionally, Sup35 has been shown to accelerate the aggregation of tau and alpha-synuclein proteins in Alzheimer's and Parkinson's disease models, respectively (Flach et al. AD 2022; Meng et al. Sci. Adv. 2023). These effects were demonstrated via intrahippocampal inoculation of Sup35NM fibrils in P301S tau transgenic mice and nasal infection with *Saccharomyces cerevisiae* in α -syn A53T transgenic mice.”*

3) The key ideas of the study, including the identification of theoretically prionogenic sequences in the microbiome, their link to human diseases e.g. *Helicobacter pylori* infections (a frequent and relevant disease) and their toxicity remain difficult to be found in the exhausting lengths and wide complexity of the paper. Although the plethora of approaches and ideas is impressive, reducing the data set to the most relevant sequences and providing the other data in Suppl Data would help the understanding.

We sincerely thank the reviewer for their valuable feedback and appreciate the recognition of the wide range of approaches and ideas presented in our study. We understand the concern regarding the complexity and length of the manuscript. To address this, we have worked to highlight the main ideas and reduce the overall main text length by moving some technical explanations to the Methods section.

While we understand the suggestion to focus on a subset of sequences for clarity, we believe this would not fully represent the broader scope and significance of our findings. Our study aims to highlight the diversity and complexity of exogenous prion-like proteins, even beyond the microbiome, by analyzing not only their theoretical prionogenic potential but also their diverse biological effects.

It is true that *Helicobacter pylori* was prominently identified in the computational analysis and that its proteins showed consistent results across all experiments, including being among the most toxic and propagation-prone sequences. However, our primary objective was not to center the work around *H. pylori* but to demonstrate the sequence and effect diversity of exogenous prion-like proteins. Retaining sequences with lower effects in the main text rather than relegating them to supplementary material is essential to show this diversity.

Some sequences may not present strong effects *in vitro* or during yeast propagation but exhibit significant toxicity *in vivo*. Moreover, this broad dataset allowed us to conduct analyses that provided valuable insights into potential toxic mechanisms. For example, while HP1 and HP2 exhibit the strongest toxicity *in vitro*, other sequences such as RI5, RI6, and HA10 result in pronounced cognitive decline *in vivo*. This broader perspective underscores the multiple ways through which these sequences might influence human health.

We hope this explanation clarifies our decision to present the complete dataset, which we believe strengthens the overall impact and relevance of the study.

4) The connection of the observed effects in *C. elegans* to the amyloid-toxicity are difficult to understand and the proposed mechanism (lipid oxidation) does not convince me. I could easily live without the *C. elegans* part of the manuscript. The manuscript should then focus on the Amyloid-interaction of the identified sequences.

We appreciate the reviewer's thoughtful feedback regarding the *C. elegans* experiments. However, we believe the *C. elegans* model provides valuable insights into the biological relevance of exogenous prion-like proteins in a living organism. While the proposed mechanism of lipid oxidation may require further investigation, the observed effects in *C. elegans* are consistent with results obtained in cell culture, supporting the potential impact of these prion-like sequences beyond their *in vitro* interactions. These *in vivo* findings emphasize the physiological significance of our results and point to pathways worth exploring in future research.

Therefore, we respectfully propose to retain the *C. elegans* section in the manuscript. To address the reviewer's concerns, we have revised the text to reduce its complexity improving the fluidity and connection between the different experiments and highlighting the key ideas.

Minor Comments:

1) Figure 2A : Some TEM figures hardy allow to recognize fibrils.

We have revised the figures and repeated the assay when required to obtain better images.

2) Figure 3A : The differences in lag-times seem minor. The assay lacks a negative (inhibitory) control.

We thank the reviewer for their observation regarding Figure 3A. We agree that the differences in lag-times may appear minor, despite they are statistically significant. To better illustrate these differences, we have modified the x-axis of Figure 3A to display the first 10 hours of the aggregation kinetics. This adjustment amplifies the distinction between the lag-times and provides a clearer visualization of the observed effects. We have also revised the figure legend and the corresponding text in the manuscript to emphasize the significance of these differences and include the statistical analysis to underline the relevance of the observed changes.

Regarding the inclusion of a negative (inhibitory) control, we appreciate the suggestion. While we did not include a conventional negative control in this particular assay, the peptide HA10 already demonstrates a reduction in aggregation compared to other conditions, serving as an inhibitory reference.

3) The identified sequences should be labelled according to their "logical" origin e.g. *Helicobacter pylori* "HP1" etc. *Coprococcus comes* "CC1"....

We thank the reviewer for their suggestion regarding the labeling of the identified sequences. We understand the importance of clearly linking the sequences to their species of origin.

In our current manuscript, we have labeled the sequences as "C" (standing for amyloid Core) followed by a number (e.g., C1, C2) to reflect their ranking based on the pWALTZ score, which is a key measure that correlates well with the results from our experiments. This approach allows for a consistent and logical arrangement of the samples across various figures and graphs, ensuring clarity in the presentation of the data.

However, as the reviewer suggested, we recognize the value of connecting the labels to their species of origin. To address this, we propose keeping the numeric order based on pWALTZ scores (as it is crucial for the consistency of our analysis) but modifying the prefix to reflect the species. For example:

"HP" for *Helicobacter pylori* sequences, resulting in HP1 for C1.

"CC" for *Coprococcus comes* sequences, resulting in CC3 for C3.

This hybrid approach would maintain the logical and consistent organization based on the pWALTZ score while also providing a clear link to the species of origin. We believe this will improve the clarity and accessibility of the data without sacrificing the consistency of our results. The final names can be consulted in Table 1:

Proportion Consensus Position																						Uniprot	pW	Protein description	Species
	1	2	3	4	5	6	7	8	9	10	11	12	13	14	15	16	17	18	19	20	21				
HP1	G	N	N	N	N	S	V	I	S	F	N	Q	T	N	F	N	Q	G	T	Y	N	M5YJZ4	78,866	putative vacuolating cytotoxin	Helicobacter pylori
HP2	S	S	V	Y	W	L	N	S	V	N	E	N	N	N	N	K	S	Y	Y	I	S	M3SIJ9	75,856	putative vacuolating cytotoxin	Helicobacter pylori
CC3	M	Q	A	A	F	N	F	I	N	N	R	R	Y	Q	D	A	I	N	V	L	N	C0B555	78,118	DnaJ domain protein	Coprococcus comes
BH4	Y	Q	Q	A	L	N	V	L	S	R	I	Q	N	R	N	A	Q	W	Y	Y	L	C9L6N5	75,187	DnaJ domain protein	Blautia hansenii
RI5	H	A	N	E	G	N	V	N	Q	Y	N	N	S	Y	Q	N	T	N	S	Y	S	D4KZ46	77,242	trypsin-like serine proteases	Roseburia intestinalis
RI6	M	D	N	N	Y	Y	N	N	N	N	N	N	N	T	N	S	N	S	N	Y	N	C0FUS6	77,13	trypsin	Roseburia inulinivorans
SA7	R	S	G	N	Y	V	F	L	A	N	S	T	N	R	Y	Q	G	N	Y	Y	N	E7GX55*	75,713	penicillin-binding protein 1A	Streptococcus anginosus
LD8	S	Y	S	S	S	A	N	N	Y	G	Y	G	S	N	Y	N	Y	S	Y	S	A	F0HUU2	74,582	aggregation promoting factor	Lactobacillus delbrueckii
P9	G	Y	V	Q	S	S	Y	G	Q	N	Q	F	Q	N	S	Q	Y	G	S	Y	Q	F5LF86	73,865	putative RNA-binding protein FUS	Paenibacillus
HA10	S	T	G	Y	S	D	L	N	S	T	Y	N	Q	S	A	L	I	N	Q	I	G	G9Y7N7	73,722	curlin associated repeat	Hafnia alvei

4) The abstract should focus on the relevant logic of the manuscript, e.g. the identification of *Helicobacter pylori* sequences, their co-aggregation effects with Abeat, and their effect on neuronal cells, including the limitation of the study, by using artificial sequence fragments without the presence of corresponding proteins in nature.

We have revised the abstract highlighting the identification of *H. pylori* sequences and indicating that in this study we use a computational approach to design synthetic sequences. There is a copy of the new abstract in answer to Comment 1

Reviewer #2:

In this study, Curto et al. investigated the biophysical and pathogenesis properties of ten gut microbe prion-like amyloid-forming core sequences. First, the authors identified prion-like sequences from gut bacteria through bioinformatic analysis. Then, they analyzed the aggregation properties using transmission electron microscopy, Fourier-transform infrared, spectroscopy, and binding to the amyloid dyes thioflavin-T and Congo red, and found that these core peptide sequences can self-assemble to amyloid fibrils and promote soluble A β 40 aggregation in vitro. Finally, Curto et al. tested the potential pathological role in cell culture, yeast and *C. elegans* and found that all peptides tested promote aggregation and display toxic effects. The finding of pathogenic roles of exogenous amyloid-forming core peptides is really interesting and consistent with Wang's reports regarding the effect of bacterial curli amyloid fibril on *C. elegans* published in 2021(PNAS, 118(34)). However, the data is preliminary. Some experiments require better controls and the mechanisms underlying the toxicity of these peptides have not been addressed. And it would be more interesting to know if the same effects exist in vertebrates.

We sincerely thank the reviewer for their detailed and thoughtful summary of our study. We are pleased that they found the discovery of pathogenic roles for exogenous amyloid-forming core peptides to be both interesting and aligned with recent literature. We appreciate their recognition of the relevance of our findings and their connection to broader research in the field.

In response to the reviewer's constructive feedback, we have revised our manuscript to include additional controls and expanded our discussion to better address the mechanisms underlying the observed toxicity. Specifically, we have highlighted oxidative stress as a potential mechanism driving the toxic effects observed in our models. Furthermore, we discuss how the maturity of the aggregated structures may influence their interaction with host cells and contribute to the observed effects. We also wish to share that we are actively pursuing a follow-up project to decipher the mechanisms of toxicity in the *C. elegans* model. We hope these revisions enhance the clarity and robustness of our work and demonstrate our commitment to further exploring the connections between exogenous amyloids and host health. We look forward to the reviewer's evaluation of these improvements.

Major points

1. In this study, the authors analyzed the aggregation and pathogenic function of 10 amyloid-forming core sequences. However, no negative control is included. Therefore, it is hard to exclude the possibility of artificial effects. I strongly recommend the authors add several negative controls (around 5 random sequences with the same length as the amyloid-forming core sequence).

We thank the reviewer for their suggestion regarding the inclusion of negative controls. We agree that it is essential to incorporate such controls to rule out the possibility of artificial effects.

We designed the control sequences using two distinct strategies. The first strategy utilized amyloid cores with lower pWALTZ scores, while the second used the amino acid composition of proteins collected in Swiss-Prot.

For the first approach, we start by randomizing sequences with lower pWALTZ scores within the positive set. To ensure low amyloidogenic and prion-forming propensity, we implemented the following modifications:

- Substituting hydrophobic residues with glycine, an amino acid with very low prion propensity (Sabate et al, 2015).
- Maintaining a Q/N content between 9.5% and 38% while ensuring a net charge below 3, to avoid repulsive forces that might artificially inhibit aggregation and to preserve some sequence similarity with the prion-positive set.

The resulting sequences were as follows:

```
>L1
GYHEGQGGYHDGGQHGGYHGG
>L2
SGHGSHESSGGSQQSRGSGGTS
>L3
NSMERNSSNSSGHGNNQDSNN
>L4
KSDQQGSHEQGMQSGMGDGMT
>L5
NNGESGGNNSGGSNNDNTSSQ
```

While this set of sequences was specifically designed to have low prion-like and aggregation propensity, it does not represent random sequences as suggested by the reviewer. Therefore, we developed a second set of control sequences using a Python script (see below) to generate random sequences of 21 residues with an amino acid composition similar to natural proteins based on Swiss-Prot data (<https://web.expasy.org/docs/relnotes/relstat.html>).

The sequences generated were:

```
>R1
GELAMARRGNPTLGGITRKRS
>R2
IDWRANVTDEPQVAAGAHTVE
>R3
KAEHTDTLMQEGARRTTDNQG
```

Python script used:

```
import random

# Define the amino acids and their frequencies
amino_acids = [
    ('A', 8.25), ('C', 1.38), ('D', 5.46), ('E', 6.71),
    ('F', 3.86), ('G', 7.07), ('H', 2.27), ('I', 5.91),
    ('K', 5.80), ('L', 9.64), ('M', 2.41), ('N', 4.06),
    ('P', 4.74), ('Q', 3.93), ('R', 5.52), ('S', 6.65),
    ('T', 5.36), ('V', 6.85), ('W', 1.10), ('Y', 2.92)
]

# Create a list of amino acids according to their frequency
aa_list = []
for aa, freq in amino_acids:
    aa_list.extend([aa] * int(freq * 100)) # Scale up for better
distribution

# Function to generate a random protein sequence of a given
length
def generate_protein_sequence(length):
    return ''.join(random.choice(aa_list) for _ in
range(length))

# Generate a random sequence of 21 residues
sequence = generate_protein_sequence(21)
print(sequence)
```

Using this approach, we obtained eight different sequences of 21 amino acids. These sequences showed negative results for aggregation and prion propensity when tested with various algorithms used in this study (PAPA, PLAAC, pWALTZ) as well as Aggrescan (Conchillo-Solé et al, 2007).

Experimental observations using TEM, Thioflavin-T, and Congo Red confirmed that these sequences did not self-assemble into amyloid-like aggregates. Consequently, they were not tested in the seeding assays (see Appendix Figures S3, S4, S5, S11, S13, S15).

When incorporated as replacements for the nucleation domain of Sup35-GFP, the resultant chimeras failed to form intracellular aggregates in yeast, except for L2, where 13% of cells exhibited foci (lower than the 61% to 100% obtained with the prion-like sequences).

No extracellular aggregates were observed when these chimeras were expressed in the *E. coli* C-DAG system.

Furthermore, worms fed *E. coli* expressing L1, L2, or R1 exhibited no significant learning or memory deficits.

Overall, these results confirm that the control sequences have low amyloidogenic propensity and fail to aggregate both *in vitro* and *in vivo*. Moreover, the absence of cognitive decline in *C. elegans* suggests that the aggregation behavior encoded in the nucleation domain of Sup35 may be linked to its toxicity. These findings reinforce the conclusion that the aggregation and propagation abilities of amyloidogenic sequences are critical for their toxic effects.

2. Fig.2, what is the concentration for the experiment? Have the experiments been replicated (for B and D)? If not, please do repeat.

We thank the reviewer for this observation. The preparation of the peptide aggregates is not well explained. A new explanation has been added to the Methods section.

“The best results were obtained by redissolving the peptides with DMSO to maintain them as monomers and diluting them in 50 mM phosphate buffer (PB) pH 7.4 at a final concentration ranging from 50 to 400 μ M and left overnight for aggregation (see Table EV2).”

The exact final concentration used for each peptide has been reported in Table EV2.

For figures B and D three biological replicates with three technical replicates were performed. The sentence below was added for clarity:

“All experiments were conducted with three biological replicates, each with three technical replicates.”

3. Fig.3, In cell culture, the aggregation properties of the prion-like core peptides are not correlated with the ROS or cytotoxicity. What is the role of the aggregation in the pathogenesis?

We agree with the reviewer that the manuscript would benefit from a discussion linking the cell culture findings with the aggregates' implications in pathogenesis. In this sense, our analysis of the cell culture assay primarily found a correlation with the net charge of the peptides (old Supplementary Figure 10). Taking into consideration this result and the reviewer's indication, we have revised the text to discuss the effect that the electrostatic forces and the aggregation of these peptides might have on pathogenesis.

We also moved from the supplementary material to the main text the plot showing the correlation with the net charge and the cell viability and added a figure showing how the aggregation properties observed can play a role in pathogenesis.

In the main text:

“The MTT assays showed that peptides with a net charge close to 0 exhibited greater toxicity compared to those with a higher charge (Fig 3G). This suggests that electrostatic interactions might play a protective role against cytotoxicity. While differing from the A β results, where positively charged peptides accelerated the aggregation kinetics (Fig 3D), this also indicates the existence of additional mechanisms contributing to toxicity.

Polar, uncharged polypeptides tend to exhibit amphipathic characteristics that facilitate their interaction with membranes. Similar to antimicrobial peptides, these interactions can lead to membrane disruption and subsequent cell toxicity. The observed general drop in cell viability to below 70% (Figure 3E) suggests that the selected amyloid cores from the gut microbiome can induce cellular damage upon aggregation.”

“Overall, our results suggest that prion-like aggregates can cause cellular damage through membrane interactions and ROS production(De Groot & Burgas, 2015; Auten & Davis, 2009). This supports their potential to contribute to diseases such as Alzheimer’s and Parkinson’s, where the accumulation of amyloid aggregates is a hallmark of pathogenesis (Fig 3H).”

4.Fig.4, all chimeric Sup35s aggregate, which can't exclude the possibility that a random sequence in the N-terminal could induce the aggregation. A negative control is required to exclude the possibility.

We agree with the reviewer that including a negative control is essential to rule out the possibility of random sequences in the N-terminal inducing aggregation. To address this, we have introduced at the N-terminal of Sup35 the same negative control sequences described in our response to Comment 1 of reviewer 2. The results obtained are presented in Appendix figures S11,13,15.

5. Only one full-length protein (C9L6N5, a protein with a DNAJ domain) was tested for the role in A β 40 aggregation, which is not typical of ThT fluorescence.

We thank the reviewer for their comment regarding the testing of full-length proteins. We acknowledge the importance of exploring additional full-length proteins for their role in A β 40 aggregation. However, due to several challenges encountered during the revision period, including those inherent to the expression of disorder and aggregation prone proteins, we were unable to purify another protein.

We appreciate the reviewer’s understanding of these limitations and remain committed to exploring additional full-length proteins in future work to further strengthen our findings.

6.Statistics are recommended for Figure 2C, Figure 3E, Figure 4D, Figure 6D-E, and Fig7B-C. And correlation analysis between oxidation, gut granules and memory would be helpful.

Following the reviewer’s recommendation, we have added statistical information to the legend of Figures 2C, 3E, 4D, 6D-E, and 7B-C:

- Figure 2C legend - All samples with aggregated peptides are significantly more fluorescent than Th-T alone (p-value < 0.0001 two-tailed unpaired t-test, SD).

- Figure 3E legend - All cells incubated with aggregated peptides have significantly lower viability than the control cells (Ctrl) without peptides (unpaired two-tailed t-test, p-value < 0.05, error bars are SD, N=3).

- Figure 4D legend - All assays were performed in biological triplicates (N=3). Yeast strains expressing Sup35 chimeras with the predicted amyloid cores exhibited significantly higher [PSI⁺] levels (indicating better prion propagation) compared to the strain without a prion core (Δ NM) (unpaired two-tailed t-test, p < 0.01 for all strains, except RI6, P9, and HA10, which showed significance at p < 0.05). For more details about the yeast phenotypes, refer to Supplementary Data 5.

- Figure 6D-E legend - Comparison between memory indices and whole-body oxidation ratio measured with mFTIR (error bars are SEM; for oxidation ratio N \geq 44, the measures were taken uniformly along the body of 10 different worms). E) Comparison between memory indices and whole-body protein/lipid ratio measured with mFTIR (error bars are SEM; for protein/lipid ratio N \geq 42, the measures were taken uniformly along the body of 10 different worms).

- Figure 7B-C legend - B) Comparison of lipid oxidation within the intestinal section and the number of gut granules (N=10 different worms). Groups in the red and green zones (RI6, BH4, RI5, NM vs. SA7, CC3, LD8, P9, HP2, Δ NM, HA10, HP1) show significant differences in lipid oxidation (p < 0.0001) and the number of granules (p < 0.0001) as determined by unpaired two-tailed t-test. Error bars represent the standard error of the mean (SEM). C) Comparison of memory index (N=3, biological replicates) and the number of gut granules (N=10 different worms). Groups in the red and green zones (RI6, RI5, HA10 vs. P9, SA7, NM, LD8, HP1, HP2, CC3, BH4, Δ NM) exhibit significant differences in memory (p = 0.0001) and the number of granules (p < 0.0001) as determined by unpaired two-tailed t-test. Error bars represent the standard error of the mean (SEM).

Additionally, we have included a new appendix figure (Figure S16) that provides a correlation analysis showing that the lipid oxidation and the number of granules correlates negatively with the associative memory in *C. elegans*. For the correlation between oxidation and memory, we obtained an R² value of 0.507 and a p-value of 0.021. Similarly, for the correlation between the number of granules and memory, we obtained an R² value of 0.462 and a p-value of 0.031.

Minor points

There is a mistake in "These core sequences are rich in amino acids that despite being associated with disordered conformations also keep amyloid propensities such as asparagine (N), glutamine (N), and tyrosine (Y)". The abbreviation for glutamine is Q.

The mistake has been corrected.

Reviewer #3:

The findings by S. Curto et al. suggest that ingested or exogenous prion-like protein derived from gut bacteria can form amyloids, which interfere with the aggregation of amyloid beta peptide, resulting in loss of sensory memory and increased lipid oxidation. These findings are interesting and thought-provoking as they highlight a potential link between gut microbiota as a reservoir of exogenous prion-like sequences and cognitive dysfunction. The experiments are well-planned and executed with proper controls and replicates. Their findings could be useful in other neurodegenerative diseases where the gut microbiome may play an important role. I have listed a few major and minor comments that may help to improve the current version of the manuscript.

We appreciate the encouraging feedback from Reviewer #3. We are pleased that our findings were recognized as both interesting and thought-provoking. The reviewer's positive assessment of our experimental design and execution is particularly gratifying. In response to the reviewer's comments, we have carefully addressed both the major and minor points raised, making several revisions to enhance the clarity and robustness of our manuscript. We believe that these changes improve the manuscript and align with the reviewer's suggestions, and we hope that the revised version will be viewed as a significant contribution to the field.

Major comments:

1. The authors have screened prion-like sequences using the following algorithms: PAPA, PLAAC, and pWALTZ in the gut microbiome and identify disordered sequences with a prion-like composition and provide a score that reflects the probability of a sequence behaving as a prion. I wondered if the authors compared the sequence with already reported CsgA-like proteins from the gut, as mentioned in Bhoite et al. *J Biol Chem.* 2022 (10.1016/j.jbc.2022.102088). It would be great to see any sequence similarity between the peptides identified in this work and the proteins reported by Bhoite et. al.

We thank the reviewer for their comment. The work by Bhoite et al. is indeed highly relevant to our study and aligns closely with our research interests. Following the reviewer's suggestion, we conducted a sequence alignment between CsgA and our peptides and proteins using various bioinformatics tools (T-Coffee, BLAST, and Unipro UGENE, see Figure PP1 below). Our analysis revealed no significant similarity between CsgA and any of the sequences identified in our study.

Figure PP1. Up, example of alignment with Unipro UGENE. At the left, percentages of similarity without gaps. Down, alignment between CsgA and HA10. All the BLAST alignments result in scores below 40. The sequence of uniport code of (G9Y7N7) HA10 was chosen for this figure since it was the one with higher scores of similarities.

Despite this negative result, we further analyzed the CsgA sequence using the abovementioned algorithms. Interestingly, while the PAPA algorithm yielded a score below the 0.05 threshold (-0.03), the prionW tool identified a prion domain with a score of 72.4, and the PLAAC algorithm predicted two putative prion-like regions with a log-likelihood ratio (LLR) above 10. These findings highlight the potential for prion-like sequences to exist within a wide range of diverse proteins.

Considering these analyses and the reviewer's suggestion, we have included and discussed the reference of Bhoite et al. and the diversity observed in putative prion-like proteins. Specifically:

In the introduction:

“Supporting this idea, curli extracellular amyloid aggregates formed by Escherichia coli have been shown to accelerate alpha-synuclein aggregation in PD animal models27–29.

Worryingly, numerous homologs of the curli amyloid protein (CsgA) are found across the prokaryotic kingdom, each exhibiting different effects on alpha-synuclein aggregation (Bhoite et al, 2022; Fernández-Calvet et al, 2024). Despite this, the link between the pathologic phenotype, toxic mechanism, and molecular properties of amyloid proteins remains elusive."

In the discussion:

"There, the microbiota can act as a source of amyloid-promoting proteins with proven effects on host health."

"This sequential diversity is also reflected in the increasing number of bacteria amyloid-promoting proteins with reported effects on host health."

2. Authors have shown a robust correlation between peptide net charge and lag time kinetics. I am curious if authors tried to understand the nature of the interaction between peptide and A β (like electrostatic interaction, which can be done in the presence of NaCl) as authors have shown more positive charge results in faster A β 40 aggregation and more negative charge results in slower aggregation.

We appreciate the reviewer's suggestion that explaining the nature of the interaction between peptide and A β would significantly enrich the manuscript. Indeed, we have recently published a study focused on the effects of NaCl and pH on A β 40 aggregation, where at neutral pH, A β 40 carries a net charge of -3. Our findings support the role of electrostatic repulsion as a driving force in aggregation, modulated by pH and salt concentration. Specifically, we observed that increasing NaCl reduces repulsion between the negative charges, thereby accelerating aggregation, except under acidic conditions where the charges are nearly neutralized.

In the context of the present study, conducted at neutral pH and physiological salt concentration, A β 40's negative charges create repulsion against the formation of intermolecular contacts. Therefore, positively charged peptides, similar to Na⁺ ions in higher NaCl concentrations, may neutralize the negative net charge, thus promoting A β 40 assembly. On the contrary, negatively charged peptides may enhance the repulsive forces in the mixture, which is consistent with previous studies on the inhibitory effects of negatively charged molecules.

Overall, we have revised the manuscript to include this additional information, highlighting the role of electrostatic forces:

"A more positive net charge results in faster A β 40 aggregation, whereas a more negative charge results in slower aggregation. This effect may be attributed to electrostatic repulsion forces. Under the experimental conditions used (pH 7.4 and 100 mM NaCl), A β 40 carries a negative charge (-2.9). As a result, positively charged peptides may neutralize the net charge favoring A β 40 assembly, while negatively charged peptides may enhance the repulsive forces present in the mixture. This has also been observed in screening for aggregation inhibitors."

3. Authors have shown cellular ROS production in which six out of ten peptides produced ROS compared to control without aggregates. However, it might be useful to add a positive control. We have added the signal of the positive control in Figure 3F.

4. C1 and C2 exhibited higher toxicity, which are fragments of two putative vacuolating cytotoxins and assembled into oligomers upon interaction with cellular membrane. But, somehow, it is not very clear to me how the authors come to this conclusion of interaction. It might be a good idea to add this to the discussion.

We thank the reviewer for this comment, which points out that this information should be in the discussion section and requires clarification.

Our intention was to indicate that the full-length proteins' function involves interaction with host cell membranes and assembly into disrupting pores. However, we acknowledge that this explanation was unclear, lacked appropriate referencing, and contained inaccuracies. Specifically, while vacuolating cytotoxins can form oligomers in water, their pore-forming activity is primarily driven by interactions with cell membranes. Additionally, it was not clearly stated whether this assembly behavior concerns the full-length proteins or the 21-amino-acid peptides tested in our study.

To address this, we have added the appropriate reference and revised the text for greater clarity, and moved it to the discussion section:

“Among the bacteria species detected, Helicobacter pylori stood out due to the high number of sequences coding for prion-like proteins (Supplementary data 2). Additionally, many AD patients have experienced active and/or latent Helicobacter infections, pointing to a potential role in the development of the pathology. Moreover, HP1 and HP2 are fragments of two putative vacuolating cytotoxins, proteins that assemble into disrupting pores upon interaction with host cell membranes. Remarkably, despite their short length, HP1 and HP2 retain significant toxicity, pointing to a possible role in host-pathogen interactions and disease.”

5. The main forms of A β in amyloid deposits are 40 and 42. Authors have performed experiments on prion-like peptides with A β 40. It would be great if the authors could either add few experiments with A β 42 or provide their speculation in the discussion.

We appreciate the reviewer's comment and share their interest in understanding the effects of exogenous molecules on A β 42 deposition. This topic has inspired a follow-up study in which we investigate the comparative effects of A β 40 and A β 42. Our preliminary findings indicate that A β 42 and A β 40 respond similarly to the presence of prion-like peptides, with comparable impact levels on aggregation. However, A β 42 aggregates at a faster rate, complicating precise kinetic measurements and diminishing the correlation strength between peptide properties and aggregation. Nonetheless, previous screenings of inhibitory peptides on A β 42 aggregation have reported similar findings. Specifically, the presence of molecules or peptides with a negative charge (up to a certain threshold) can reduce A β 42 aggregation.

Although we were unable to establish a detailed correlation for A β 42 under our experimental conditions, we agree with the reviewer that a discussion of this isoform is necessary. To address this, we have revised the manuscript to incorporate these insights and cite relevant literature:

“Moreover, these works also report that these electrostatic forces work similarly for A β 42 (the main isoform located in the amyloid plaques), which despite having two extra amino acids has the same net charge at physiological conditions.”

Minor comments:

1. Table 1: Last line, s is missing in species.

An s has been added at the end of the legend.

2. In the result section Ingestion of exogenous prion-like proteins increases lipid oxidation - 2nd line from the last "with" is written two times.

The **with** has been removed.

3. 3rd line in the 3rd paragraph (Without ab40 and with peptide and without ab 40 or peptide) is repeated in the Thioflavin T binding and aggregation kinetics section of material and methods.

The sentence was corrected for clarity.

“Control experiments were conducted under the same conditions, including controls without A β 40 but with peptide and with just buffer and Th-T.”

4. In nonsense suppression assay of material and method section, 2 is written in both numeric and word in the sentence.

The numerical 2 was removed.

5. Figure 2: Absorvance to be replaced by Absorbance, what is the control in CR plot?

The name of the axis has been corrected and the composition of the Congo Red control has been explained.

In the figure legend, this text has been added: *“Control is buffer with CR, without aggregates.”*

In the method section, the text has been clarified: *“Each spectrum was compared with that of CR alone (without peptide, as shown in Figure 2).”*

6. Bacterial species are to be written in italics throughout the text.

We have carefully reviewed the manuscript and ensured that all bacterial species names are now correctly italicized throughout the text.

11th Mar 2025

Manuscript Number: MSB-2024-12399R

Title: Exogenous Prion-Like Proteins and Their Potential to Trigger Cognitive Dysfunction

Dear Dr Sánchez de Groot,

Thank you for the submission of your revised manuscript to Molecular Systems Biology. I am pleased to inform you that we will be able to accept your manuscript pending the following final amendments and appropriate response to reviewers:

- 1) In line with comments from Reviewer 2, we would encourage some of the new negative control experiments with scrambled peptide sequence to be carried out in the same experiment as the amyloid-forming sequences that you have identified. If you wish to discuss this further, please feel free to write me an email or we can discuss over a phone/video call.
- 2) Please ensure that as a co-corresponding author, the ORCID ID of Dr. María Rosario Fernandez Gallegos is linked to their eJP account. A link to connect the ORCID account was previously sent to Dr. Fernandez Gallegos via email. Please also ensure that the name in our submission system matches the name in the manuscript, as currently it does not.
- 3) Please format the Data availability section according to the example below:
"The datasets and computer code produced in this study are available in the following databases:
 - Chip-Seq data: Gene Expression Omnibus GSE46748 (<https://www.ncbi.nlm.nih.gov/geo/query/acc.cgi?acc=GSE46748>)
 - Modeling computer scripts: GitHub (<https://github.com/SysBioChalmers/GECKO/releases/tag/v1.0>)
 - [data type]: [full name of the resource] [accession number/identifier] ([doi or URL or identifiers.org/DATABASE:ACCESSION])"
- 4) Please remove the section on "Relevant web links" and include these links at the point of mention in the main manuscript text.
- 5) Please rename "Competing Interest" to "Disclosure and competing interests statement". We updated our journal's competing interests policy in January 2022 and request authors to consider both actual and perceived competing interests. Please review the policy <https://www.embopress.org/competing-interests> and update your competing interests if necessary.
- 6) In the Methods, please take care of the following:
 - The Materials and Methods section should be renamed to "Methods".
 - Cell lines: Please be sure to include a sentence in the Methods as to whether or not the SH-SY5Y cell line was recently authenticated and tested for mycoplasma contamination. Please also be sure to update the Author Checklist with this information and where it can be found in the manuscript.
 - Please ensure that a statement on whether or not blinding was done is included in the Methods even if no blinding was done. Please also be sure to update the Author Checklist with this information and where it can be found in the manuscript.
- 7) Please remove the Reagents and Tools Table from the Methods section of the manuscript and upload it as a separate file choosing the file type "Reagent Table".
- 8) Please place individual sections of the manuscript in the following order: Title page - Abstract & Keywords - Introduction - Results - Discussion - Methods - Data Availability - Acknowledgements - Disclosure and Competing Interests Statement - References - Figure Legends - Expanded View Figure Legends.
- 9) For the figures and figure legends, please take care of the following:
 - Please make sure to update the callouts of all figures in the main manuscript text. Currently figure callouts are missing for Fig 4C and Fig 7A.
 - Please note that the exact p values are not provided in the legends of figures 2C, 3B, C, F; 4D, 5B, 6B, 7B, C.
 - Please indicate the statistical test used for data analysis in the legends of figures 3D, G.
 - Please note that information related to n is missing in the legends of figures 3B, C
 - Please note that the error bar is not defined in the legend of figure 4D.
 - Please note that the measure of center for the error bars needs to be defined in the legends of figures 3E, F; 5B
- 10) Tables: Table EV1 should be renamed Dataset EV1 the numbering of the subsequent EV tables should be adjusted accordingly. The legend for Dataset EV1 should be removed from the main manuscript and added to the Dataset file in a separate tab. Please also be sure to update all callouts in main manuscript text. Can Table EV3 be reformatted to make it easier to read? All the remaining Table EVX files should have their legends removed from the manuscript text and added to the excel files, including the dataset/table name displayed.
- 11) Appendix file: Please upload the Appendix as a single PDF, i.e. please add the two appendix tables as "Appendix Table S1" and "Appendix Table S2", and the file with the supplementary data should also be added to the PDF as "Appendix Supplementary Data"
- 12) Funding: Please note that funding information should be given in the "Acknowledgements" section (not in its own separate section). Please ensure that all funding sources entered into the manuscript submission system are also listed in the manuscript - currently FPU21/03897 is listed in our system but not in the manuscript text.
- 13) Synopsis:
 - Synopsis image: Please upload the synopsis image as a high-resolution jpeg or png file 550 pixels wide x (300-600) pixels high.
 - Synopsis text: Please provide a short standfirst (maximum of 300 characters, including space), limit the bullet points to max. 5 and upload it as a separate .doc file. Please write the bullet points to summarise the key NEW findings. They should be designed to be complementary to the abstract - i.e. not repeat the same text. We encourage inclusion of key acronyms and

quantitative information (maximum of 30 words / bullet point). Please use the passive voice.

14) Source Data: Please ensure that a completed Source Data checklist is uploaded as a Related Manuscript File.

15) As part of the EMBO Publications transparent editorial process initiative (see our policy here:

https://www.embopress.org/transparent-process#Review_Process), Molecular Systems Biology will publish online a Peer Review File (PRF) to accompany accepted manuscripts. This file will be published in conjunction with your paper and will include the anonymous referee reports, your point-by-point response and all pertinent correspondence relating to the manuscript. Let us know whether you agree with the publication of the PRF and as here, if you want to remove or not any figures from it prior to publication. Please note that the Authors checklist will be published at the end of the PRF.

16) After your paper is published, we will promote it on social media. If you have any handles or hashtags for Bluesky you would like included, please let us know.

17) Please provide a point-by-point letter INCLUDING my comments as well as the reviewer's reports and your detailed responses (as Word file).

I look forward to reading a new revised version of your manuscript as soon as possible.

Yours sincerely,

Poonam Bheda, PhD
Scientific Editor
Molecular Systems Biology

Reviewer #1:

The manuscript has been substantially revised and all my previous critics have been addressed in detail.

Reviewer #2:

The revised manuscript has addressed most of my comments. However, one of the main concerns regarding the negative control was addressed in an unconvincing way. As controls, they should be included in the analysis along with those 10 selected amyloid-forming sequences. Instead, they only did some analysis with those negative controls and they were done separately. Furthermore, the toxicity and oxidative effect of negative controls should be tested as well because this data will tell us whether net charge or aggregation property plays more important roles in toxicity. I also had a hard time to understand the correlation between the aggregation score and the ROS/oxidation, toxicity or cognitive deficit. For example, it is confusing to see that peptides with significant different aggregation and ROS generation properties show no significant toxicity (Fig 3B, C, F, E). And the effect of those peptides is not consistent in Fig. 6D and 6E. Specifically, the effect of HP1, HP2 NM, SA7 and P5 on memory is decreased in D but increased in E. It will be great to summarize those in a model. Some other comments/suggestions are listed as below:

1. When compare among three or more samples, t-test is not a proper statistic method. The authors should choose an appropriate statistical method.
2. The authors should list the pWALTZ score for all peptides used in the paper.
3. Fig 2A-B, no control; 2C, no control, no statistics. Fig 3E, 4D no statistics
4. In Fig 7, A and B are absent.
5. Appendix Fig S4, the AS7 should be SA7
6. Appendix fig S14 , the set of comparison for learning index is unreasonable. The N2+ Δ NM should be compared with N2+NM and CL2355+ Δ NM. Similarly, the CL2355+ Δ NM should be compared with CL2355+NM and N2+ Δ NM. The same for memory.
7. In page 9, the authors' conclusion "This suggests a potential decrease in their learning capability due toand their extracellular aggregates" is not convincing because the correlation is not obvious.
8. There are many grammar and wording issues. For example, in the abstract, in "When *C. elegans* were fed a bacterial....., they lost sensory memory.....", the "a bacterial" should be "bacteria" and "lost memory" is overstating the effect of those peptides. "*C. elegans*" and "*E. coli*" in figure legends should be italic. Please double check and correct them.

Dear Editor and Reviewers,

We thank you very much for your time, effort, and constructive feedback on our manuscript. We have carefully considered all comments and suggestions, and we believe that the revised version addresses all concerns raised.

The main changes to highlight in the revised manuscript include the addition of a new author, an updated version of Figure 2 incorporating the changes suggested by Reviewer #2, and a new Dataset EV4 that provides the previously missing p-values. The inclusion of this dataset has shifted the numbering of the subsequent Dataset EV files. The Appendix has been revised to include new figures presenting the repeated assays conducted between the first and second versions of the manuscript. Additionally, the main text has been updated to address all comments from the editorial team and reviewers, including clarifications, statistical information, and formatting corrections.

Below, we provide a detailed point-by-point response to each comment from the reviewers and the editorial team.

Manuscript Number: MSB-2024-12399R

Title: Exogenous Prion-Like Proteins and Their Potential to Trigger Cognitive Dysfunction

Dear Dr Sánchez de Groot,

Thank you for the submission of your revised manuscript to Molecular Systems Biology. I am pleased to inform you that we will be able to accept your manuscript pending the following final amendments and appropriate response to reviewer 2:

1) In line with comments from Reviewer 2, we would encourage some of the new negative control experiments with scrambled peptide sequence to be carried out in the same experiment as the amyloid-forming sequences that you have identified. If you wish to discuss this further, please feel free to write me an email or we can discuss over a phone/video call.

We thank the editor for the opportunity to clarify our experimental design in response to Reviewer #2's comment.

We regret not making it sufficiently clear in the revised manuscript that internal controls were consistently included during the new experiments with the low-aggregation-prone peptide set. These controls were incorporated specifically to ensure that data generated in separate experimental runs remained comparable and interpretable.

In addition to standard positive and negative controls, we also included previously characterized amyloid-forming peptides from our original dataset as internal reference points. These repeated samples served as experimental anchors, demonstrating consistent behavior across assays and supporting reliable comparison with our earlier results.

We have now revised the figures and figure legends related to the low-aggregation-prone peptides to explicitly indicate the inclusion of these internal controls. For clarity, we summarize the controls used in each set of experiments below:

1. Amyloid Formation In Vitro

Controls: Buffer-only and buffer + dye blanks (for both ThT and Congo Red assays)

Internal reference: AS7, a peptide shown to form amyloid fibrils in our initial experiments, included as a control across ThT fluorescence, Congo Red binding, and transmission electron microscopy (TEM) assays

2. Yeast Sup35 Aggregation (Foci Formation Assay)

Controls: Sup35 Δ NM (negative) and Sup35NM (positive) *S. cerevisiae* strains

3. Bacterial Sup35 Aggregation (CDAG System)

Controls: Sup35 Δ NM (negative) and Sup35NM (positive) *E. coli* strains

4. *C. elegans* Behavioral Assays

Controls: N2 worms fed with Sup35 Δ NM (negative) or Sup35NM (positive) *E. coli* strains

We have also provided a detailed explanation in our point-by-point response to Reviewer #2, outlining the rationale behind our control strategy and why additional toxicity assays for low-aggregation peptides were not pursued. We trust this addresses the editorial concerns and demonstrates the rigor and reproducibility of our experimental approach.

2) Please ensure that as a co-corresponding author, the ORCID ID of Dr. María Rosario Fernandez Gallegos is linked to their eJP account. A link to connect the ORCID account was previously sent to Dr. Fernandez Gallegos via email. Please also ensure that the name in our submission system matches the name in the manuscript, as currently it does not.

Thank you for your message. We have now informed Dr. María Rosario Fernandez Gallegos to link her ORCID ID to her eJP account using the email previously provided. We will also ensure that her name in the submission system matches exactly as it appears in the manuscript.

3) Please format the Data availability section according to the example below:
"The datasets and computer code produced in this study are available in the following databases:

- Chip-Seq data: Gene Expression Omnibus GSE46748
(<https://www.ncbi.nlm.nih.gov/geo/query/acc.cgi?acc=GSE46748>)

- Modeling computer scripts: GitHub
(<https://github.com/SysBioChalmers/GECKO/releases/tag/v1.0>)

- [data type]: [full name of the resource] [accession number/identifier] ([doi or URL or identifiers.org/DATABASE:ACCESSION])"

The Data Availability section has been reformatted according to the requested example, specifying data types, resources, and links.

4) Please remove the section on "Relevant web links" and include these links at the point of mention in the main manuscript text.

The "Relevant web links" section has been removed, and the corresponding links have been integrated into the main text at the appropriate locations

5) Please rename "Competing Interest" to "Disclosure and competing interests statement". We updated our journal's competing interests policy in January 2022 and request authors to consider both actual and perceived competing interests. Please review the policy <https://www.embopress.org/competing-interests> and update your competing interests if necessary.

The section has been renamed to "Disclosure and competing interests statement," and we have reviewed the policy to ensure compliance. No updates to the declared interests were necessary.

6) In the Methods, please take care of the following:

- The Materials and Methods section should be renamed to "Methods".
- Cell lines: Please be sure to include a sentence in the Methods as to whether or not the SH-SY5Y cell line was recently authenticated and tested for mycoplasma contamination. Please also be sure to update the Author Checklist with this information and where it can be found in the manuscript.
- Please ensure that a statement on whether or not blinding was done is included in the Methods even if no blinding was done. Please also be sure to update the Author Checklist with this information and where it can be found in the manuscript.

The section title has been changed from "Materials and Methods" to "Methods."

A sentence confirming authentication and mycoplasma testing of the SH-SY5Y cell line has been added.

A statement indicating that partial blinding was applied has been included.

Finally, these updates have been reflected in the Author Checklist as well.

7) Please remove the Reagents and Tools Table from the Methods section of the manuscript and upload it as a separate file choosing the file type "Reagent Table".

The Reagents and Tools Table has been removed from the manuscript and uploaded separately under the file type "Reagent Table."

8) Please place individual sections of the manuscript in the following order: Title page - Abstract & Keywords - Introduction - Results - Discussion - Methods - Data Availability - Acknowledgements - Disclosure and Competing Interests Statement - References - Figure Legends - Expanded View Figure Legends.

The manuscript has been reorganized to follow the required order: Title page – Abstract & Keywords – Introduction – Results – Discussion – Methods – Data Availability – Acknowledgements – Disclosure and Competing Interests Statement – References – Figure Legends – Expanded View Figure Legends.

9) For the figures and figure legends, please take care of the following:

- Please make sure to update the callouts of all figures in the main manuscript text. Currently figure callouts are missing for Fig 4C and Fig 7A.

Callouts for Fig 4C and Fig 7A have been added in the main manuscript text

- Please note that the exact p values are not provided in the legends of figures 2C, 3B, C, F; 4D, 5B, 6B, 7B, C.

The exact p values have been indicated in the legends of figures 7B, 7C and in Dataset EV4 for 2C, 3B, 3C, 3F; 4D, 5B, 6B.

- Please indicate the statistical test used for data analysis in the legends of figures 3D, G.

The statistical test used for data analysis has been indicated in the legends of figures 3D, G.

- Please note that information related to n is missing in the legends of figures 3B, C

The number of replicates has been indicated in the legends of figures 3B, C.

- Please note that the error bar is not defined in the legend of figure 4D.

The error bar has been defined in the legend of figure 4D.

- Please note that the measure of center for the error bars needs to be defined in the legends of figures 3E, F; 5B

The measure of center for the error bars has been defined in the legends of figures 3E, F; 5B

10) Tables: Table EV1 should be renamed Dataset EV1 the numbering of the subsequent EV tables should be adjusted accordingly. The legend for Dataset EV1 should be removed from the main manuscript and added to the Dataset file in a separate tab. Please also be sure to update all callouts in main manuscript text. Can Table EV3 be reformatted to make it easier to read? All the remaining Table EVX files should have their legends removed from the manuscript text and added to the excel files, including the dataset/table name displayed.

All Tables EV haven been renamed and formatted as requested. Their legends have been removed from the main manuscript and added to the dataset file in a separate tab. All in-text callouts have been corrected. Table EV3 has also been reformatted to improve readability.

11) Appendix file: Please upload the Appendix as a single PDF, i.e. please add the two appendix tables as "Appendix Table S1" and "Appendix Table S2", and the file with the supplementary data should also be added to the PDF as "Appendix Supplementary Data"

The Appendix has been uploaded as a single PDF file. An "Appendix Supplementary Data" section has been added at the end of the Appendix file incorporating the old Appendix Supplementary Data and both Appendix Table S.

12) Funding: Please note that funding information should be given in the "Acknowledgements" section (not in its own separate section). Please ensure that all funding sources entered into the manuscript submission system are also listed in the manuscript - currently FPU21/03897 is listed in our system but not in the manuscript text.

The funding information has been moved to the Acknowledgements section, and all funding sources listed in the submission system, including FPU21/03897, have been added to the manuscript text.

13) Synopsis:
- Synopsis image: Please upload the synopsis image as a high-resolution jpeg or png file 550 pixels wide x (300-600) pixels high.
- Synopsis text: Please provide a short standfirst (maximum of 300 characters, including space), limit the bullet points to max. 5 and upload it as a separate .doc file. Please write the bullet points to summarise the key NEW findings. They should be designed to be complementary to the abstract - i.e. not repeat the same text. We encourage inclusion of key acronyms and quantitative information (maximum of 30 words / bullet point). Please use the passive voice.
- Please check your synopsis text and image before submission with your revised manuscript. Please be aware that in the proof stage minor corrections only are allowed (e.g., typos).

The synopsis image has been uploaded as a high-resolution JPEG file, and the synopsis text has been provided in a separate .doc file, following all formatting requirements.

14) Source Data: Please ensure that a completed Source Data checklist is uploaded as a Related Manuscript File.

The completed Source Data checklist has been uploaded as a Related Manuscript File.

15) As part of the EMBO Publications transparent editorial process initiative (see our policy here: https://www.embopress.org/transparent-process#Review_Process), Molecular Systems Biology will publish online a Peer Review File (PRF) to accompany accepted manuscripts. This file will be published in conjunction with your paper and will include the anonymous referee reports, your point-by-point response and all pertinent correspondence relating to the manuscript. Let us know whether you agree with the publication of the PRF and as here, if you want to remove or not any figures from it prior to publication. Please note that the Authors checklist will be published at the end of the PRF.

We agree with the publication of the PRF.

16) After your paper is published, we will promote it on social media. If you have any handles or hashtags for Bluesky you would like included, please let us know.

We think the following hashtags would be appropriate: #Amyloid, #Prion, #Neurodegeneration, #Microbiome, #Gut and #UABBarcelona.

17) Please provide a point-by-point letter INCLUDING my comments as well as the reviewer's reports and your detailed responses (as Word file).

A Word file including all the comments and responses have been uploaded as a point-by-point letter.

Reviewer #1:

The manuscript has been substantially revised and all my previous critics have been addressed in detail.

We sincerely thank Reviewer #1 for their positive assessment and for acknowledging the improvements made in the revised manuscript. We truly appreciate their thoughtful comments during the review process.

Reviewer #2:

The revised manuscript has addressed most of my comments. However, one of the main concerns regarding the negative control was addressed in an unconvincing way. As controls, they should be included in the analysis along with those 10 selected amyloid-forming sequences. Instead, they only did some analysis with those negative controls and they were done separately.

We sincerely appreciate the reviewer's thoughtful feedback and apologize if our previous explanation was unclear. It is important to note that all new experiments included appropriate controls (e.i., repeat samples used before) ensuring consistency with previous observations and enabling direct comparisons.

To improve clarity, we have updated the Appendix Figures to ensure that all comparable data of each experiment are presented together. Additionally, we have reanalyzed the statistical data to provide a more comprehensive comparison (see Extra Figures 1 and 2).

Extra Figure 1. New statistical analysis and control comparison for the Thioflavin-T binding assay. **A)** The plot shows the thioflavin-T fluorescence for all the samples analyzed in this work. In blue and green are indicated the repeated samples. “old” are the measures of the first version of the article and “new” those obtained during the analysis of the low aggregation prone sequences. **B)** Table showing that the significances against their corresponding blanks (ThT basal fluorescence) obtained including all the samples are similar to those reported in the first version of the article. The analysis was performed using multiple unpaired two-tailed t-tests, corrected for multiple comparisons using the Benjamini–Hochberg method (N=3, FDR set at 5%). **C)** Comparison between the same samples of the two repeated assays. The analysis was performed using multiple unpaired two-tailed t-tests, corrected for multiple comparisons using the Benjamini–Hochberg method (N=3, FDR set at 5%). In the second assay the aggregates of SA7 peptide resulted in an even higher ThT fluorescence, supporting that the conditions used were favorable for the formation of aggregates. Despite this, the low aggregation prone sequences did not achieve a fluorescence increase significantly higher than the blank.

B

Learning	q value	Significantly
NM	0,9421	No
HP1	0,0135	Lower
HP2	0,0071	Lower
CC3	0,1995	No
BH4	0,5103	No
R15	0,0109	Lower
R16	0,0089	Lower
SA7	0,4870	No
LD8	0,0991	No
P9	0,5356	No
HA10	0,0050	Lower
new_NM	0,5450	No
L1	0,3733	No
L2	0,0024	Higher
R1	0,6238	No

Memory	q value	Significantly
NM	0,0059	Lower
HP1	0,0110	Lower
HP2	0,0109	Lower
CC3	0,1075	No
BH4	0,7984	No
R15	0,0020	Lower
R16	0,0134	Lower
SA7	0,0047	Lower
LD8	0,0144	Lower
P9	0,0156	Lower
HA10	0,0072	Lower
new_NM	0,0140	Lower
L1	0,5021	No
L2	0,6608	No
R1	0,2670	No

Extra Figure 2. New statistical analysis and control comparison for the chemotaxis assay. **A)** The plots show the learning and memory indices of all the samples analyzed in this work. In blue and green are indicated the repeated samples. “old” are the measures of the first version of the article and “new” those obtained during the analysis of the low aggregation prone sequences. **B)** Tables showing that the significances against their corresponding Δ NM obtained including all the samples are similar to those reported in the first version of the article. The analysis was performed using multiple unpaired two-tailed t-tests, corrected for multiple comparisons using the Benjamini–Hochberg method ($N=3$, FDR set at 5%). **C)** Comparison between the same samples of the two repeated assays. The analysis was performed using multiple unpaired two-tailed t-tests, corrected for multiple comparisons

using the Benjamini–Hochberg method (N=3, FDR set at 5%). The learning and memory indices obtained for both samples (Δ NM and NM) in the repeated assays, present no significant differences.

Finally, we provide below a summary of the assays presented and their rationale:

1. Amyloid Formation In Vitro

- Goal: Confirm amyloid formation via Thioflavin-T (ThT) fluorescence, Congo Red binding and transmission electron microscopy (TEM).
- Outcome: The presence of amyloid fibrils was confirmed in aggregation-prone sequences, while the negative controls did not form amyloid structures (Appendix Figure S3-S5).
- Internal control: AS7 peptide.

2. Amyloid Seeding of A β 40

- Rationale: Since the low-aggregation-prone controls do not form amyloid aggregates, they do not provide material to function as seeds.

3. Toxicity & ROS Assays

- Goal: Determine whether amyloid fibrils induce toxicity and reactive oxygen species (ROS) production.
- Rationale: Since the low-aggregation-prone controls do not form amyloid aggregates, is not possible to test the toxicity of these aggregated forms.

4. Substitution of the Sup35 Prion Nucleation Region

- Outcome: Sup35 chimeras with amyloid-like sequences formed foci, whereas negative controls either did not aggregate or exhibited reduced aggregation. The last is a well-documented feature associated with a reduced or absent ability to propagate the [PSI⁺] phenotype (Meng et al, 2023; Flach et al, 2022; Burra et al, 2021; Balbirnie et al, 2001; Krishnan & Lindquist, 2005; Parham et al, 2001; Wickner et al, 2015; Sivanathan & Hochschild, 2013).
- Internal controls: Sup35 Δ NM and Sup35NM strains of *S. cerevisiae*.

5. Extracellular Amyloid Aggregate Formation in *E. coli*

- Outcome: TEM imaging confirmed no extracellular amyloid aggregates were formed when expressing the low aggregation prone controls.
- Internal controls: Sup35 Δ NM and Sup35NM strains of CDAG *E. coli* strains (REF).

6. *C. elegans* Behavioral Assays

- Outcome: 80% of aggregation-prone sequences led to significant memory decline, while at least the first three low-aggregation-prone sequences tested (40% of the total set) showed no memory defects, indicating that the proportion of sequences capable of triggering disease in *C. elegans* is lower in the low-aggregation-prone set.
- Internal controls: *C. elegans* N2 fed with C-DAG *E. coli* strains expressing Sup35 Δ NM and Sup35NM.

Furthermore, the toxicity and oxidative effect of negative controls should be tested as well because this data will tell us whether net charge or aggregation property plays more important roles in toxicity.

We sincerely appreciate the reviewer's suggestion regarding toxicity assays for the negative control peptides. However, we believe that, within the specific scope and objectives of our study, these additional assays would not substantially alter or enhance our conclusions.

Our work is focused on testing whether microbiome-derived sequences with prion-like composition can form amyloid-like aggregates and induce pathological phenotypes. The negative controls were specifically designed to lack amyloidogenic or prion-like features to demonstrate (as requested by the reviewer in the previous revision) that the ability to aggregate is not random but sequence-specific.

Indeed, the low-aggregation-prone sequences did not form fibrillar aggregates (Appendix Figure 13), supporting their function as structural and compositional controls. Given their lack of aggregation, any observed toxicity may be associated to a different property, such as an unexpected feature introduced by the sequence randomization, and would fall outside the scope of our current study.

Furthermore, the correlation between charge, aggregation and toxicity is presented in the manuscript not as a demonstration of causality, but as supporting evidence that our synthetic peptides behave consistently with known amyloid-forming sequences. Our manuscript does not claim a direct causal relationship between net charge, aggregation, and toxicity, as these aspects have been extensively analyzed in the literature (Chiti et al., 2002; Chiti & Dobson, 2006; Uversky, 2017). Instead, we use these trends to reinforce the validity of our predictive framework and experimental assays.

I also had a hard time to understand the correlation between the aggregation score and the ROS/oxidation, toxicity or cognitive deficit. For example, it is confusing to see that peptides with significant different aggregation and ROS generation properties show no significant toxicity (Fig 3B, C, F, E).

We appreciate the reviewer's concern and understand the difficulty in interpreting the correlation between aggregation scores, ROS generation, toxicity, and cognitive deficits.

First, the interference with A β aggregation (Fig. 3C and 3D) is not necessarily expected to correlate directly with cell toxicity. Instead, it indicates that, if these proteins interact, they may influence each other's aggregation behavior. However, this does not imply a direct toxic effect.

Regarding to Figure 3E, the legend states: "*All cells incubated with aggregated peptides have significantly lower viability than control cells (Ctrl) without peptides (t-test).*" The absence of p-values may have led to confusion, so we have now reported the p-values at the Appendix Table EV4.

Additionally, no direct correlation between ROS generation and toxicity levels was observed. This was explicitly noted in the main text, where we clarified that ROS production may contribute to overall toxicity but does not necessarily explain the full mechanism.

Overall, amyloid fibril toxicity can arise through multiple mechanisms, including interactions with membranes, mitochondrial dysfunction, and disturbances in protein quality control. Without in-depth biochemical analyses, such as assessing cellular stress responses or general integrity, it is difficult to reveal a definitive pathway.

And the effect of those peptides is not consistent in Fig. 6D and 6E. Specifically, the effect of HP1, HP2 NM, SA7 and P5 on memory is decreased in D but increased in E. It will be great to summarize those in a model.

We appreciate the reviewer's observation regarding the differences between Figures 6D and 6E. These differences arise from the fact that the groupings in each figure are based on distinct parameters. Specifically, in Figure 6D, samples were grouped according to memory, while in Figure 6E, they were classified based on their protein-to-lipid ratio.

During this revision, we also realized that the legend for Figure 6 did not indicate that the arrows within the plot represent the correlation shown in Figure S16. We have now updated the legend to clarify both the sample groupings and the reference to the correlation.

Regarding the suggestion to include a model, we would like to note that our hypothesis is already stated at the end of the section "The aggregates' properties define the cognitive deficit", as follows:

"Considering all factors together, we hypothesize that the inability to progress to mature and inert aggregates may lead to the formation of promiscuous conformations that generate more ROS and ultimately promote stronger cognitive decline."

Additionally, we believe that the graphical abstract already provides a visual summary of the proposed connection between aggregation, propagation, ROS generation, and cognitive impairment, and therefore an additional model figure may not be necessary.

Some other comments/suggestions are listed as below:

1. When compare among three or more samples, t-test is not a proper statistic method. The authors should choose an appropriate statistical method.

We thank the reviewer for this observation, which helped us recognize that the statistical methods were not always clearly described in the figure legends.

After a careful review, we found that in all plots, except for Figure 6B and Appendix Figure S14, we only compared each sample to its corresponding control. In those cases, statistical significance was assessed using multiple unpaired two-tailed t-tests, corrected for multiple

comparisons using the Benjamini–Hochberg False Discovery Rate (FDR) method, with FDR set at 5%.

For Appendix Figure S14, which includes comparisons across multiple groups, we used ANOVA. Additionally, in Figure 6B, comparisons between distinct conditions (marked with hash symbols) were also analyzed using ANOVA.

We have now revised the figure legends to clearly state the statistical tests used in each case.

2. The authors should list the pWALTZ score for all peptides used in the paper.

We thank the reviewer for noticing that we had omitted the pWALTZ scores for the low-aggregation-prone control peptides. However, all of these sequences yielded pWALTZ values below 50.0, which is the established threshold for distinguishing prion-like sequences (Zambrano et al, 2015). As a result, the pWALTZ algorithm applied does not provide scores below this cutoff.

To address this omission, we have now included this information in the Methods section, where we describe the design strategy for the low-aggregation-prone sequences.

3. Fig 2A-B, no control; 2C, no control, no statistics. Fig 3E, 4D no statistics.

We thank the reviewer for highlighting these points and have revised the figures and legends to improve clarity regarding controls and statistical analysis.

- Figure 2A: TEM image of the control of just buffer (Blank) was added.
- Figure 2B: IR absorbance data now include the buffer as a negative control.
- Figure 2C: Fluorescence is expressed as fold change relative to the ThT-only blank, which is now explicitly shown in the plot. While statistical markers are not displayed in the figure, the legend specifies that all aggregated peptide samples were significantly more fluorescent than the blank, with exact values now provided in Dataset EV4.
- Figure 3E: Cell viability data were analyzed using multiple unpaired two-tailed t-tests with FDR correction (Benjamini–Hochberg method, 5%). While statistical markers are not shown in the plot, the legend now states the test used, and full values are reported in Dataset EV4.
- Figure 4D: Each Sup35 chimera was compared individually to the Δ NM control to assess prion propagation. Although no significance markers are shown in the plot, the legend now reports statistical testing (FDR-corrected unpaired t-tests), and now all results are provided in Dataset EV4.

We hope these revisions and clarifications address the reviewer's concerns and improve the transparency of the data presentation.

4. In Fig 7, A and B are absent.

We have downloaded the PDF merge version and the letters from panel A and B are visible.

5. Appendix Fig S4, the AS7 should be SA7

We appreciate the reviewer's attention to detail. The labeling error has been corrected, and "AS7" has been updated to "SA7."

6. Appendix fig S14 the set of comparison for learning index is unreasonable. The N2+ Δ NM should be compared with N2+NM and CL2355+ Δ NM. Similarly, the CL2355+ Δ NM should be compared with CL2355+NM and N2+ Δ NM. The same for memory.

In our previous figure, we compared all samples against each other, and only those with significant differences were marked with stars. We have now adjusted the comparisons to align with the reviewer's suggestions.

7. In page 9, the authors' conclusion "This suggests a potential decrease in their learning capability due toand their extracellular aggregates" is not convincing because the correlation is not obvious.

We appreciate the reviewer's comment regarding the correlation between extracellular aggregates and learning impairment. We agree that the wording at the end of that paragraph could have been unclear, and to address this, we have updated the final sentence to:

"This suggests that the consumption of these bacterial strains and their extracellular aggregates has a stronger physiological impact on learning ability. Notably, the only variable among these conditions is the E. coli strain consumed and, consequently, the specific chimera it expresses."

This revised sentence strengthens our argument by explicitly clarifying the rationale behind our conclusion.

8. There are many grammar and wording issues. For example, in the abstract, in "When *C. elegans* were fed a bacterial....., they lost sensory memory.....", the "a bacterial" should be "bacteria" and "lost memory" is overstating the effect of those peptides. "*C. elegans*" and "*E. coli*" in figure legends should be italic. Please double check and correct them.

We appreciate the reviewer's attention to detail and their suggestions regarding grammar and wording. After reviewing the terminology, we recognize that sensory memory may not have been the most precise term to describe our findings. Since our study specifically assessed positive olfactory short-term associative memory using the STAM assay, we believe that referring to "lost associative memory" more accurately reflects the measured phenotype. Therefore, we have revised the sentence to:

"When C. elegans were fed bacteria expressing these prion-like proteins, they lost associative memory and exhibited increased lipid oxidation."

We appreciate the reviewer's feedback, which allowed us to refine our wording for greater clarity and scientific accuracy.

2nd May 2025

Manuscript number: MSB-2024-12399RR

Title: Exogenous Prion-Like Proteins and Their Potential to Trigger Cognitive Dysfunction

Dear Dr Sánchez de Groot,

Thank you again for sending us your revised manuscript. We are now satisfied with the modifications made and I am pleased to inform you that your paper has been accepted for publication.

Yours sincerely,

Poonam Bheda, PhD
Scientific Editor
Molecular Systems Biology
